# Inulin-Based Polymeric Micelles Functionalized with Ocular Permeation Enhancers: Improvement of Dexamethasone Permeation/Penetration through Bovine Corneas

**DOI:** 10.3390/pharmaceutics13091431

**Published:** 2021-09-09

**Authors:** Giulia Di Prima, Mariano Licciardi, Flavia Bongiovì, Giovanna Pitarresi, Gaetano Giammona

**Affiliations:** Dipartimento di Scienze e Tecnologie Biologiche, Chimiche e Farmaceutiche (STEBICEF), University of Palermo, Via Archirafi 32, 90123 Palermo, Italy; giulia.diprima@unipa.it (G.D.P.); flaviabongiovi@gmail.com (F.B.); giovanna.pitarresi@unipa.it (G.P.); gaetano.giammona@unipa.it (G.G.)

**Keywords:** ocular drug delivery, inulin, permeation enhancers, taurine, PEG_2000_, carnitine, creatine, dexamethasone, micelles

## Abstract

Ophthalmic drug delivery is still a challenge due to the protective barriers of the eye. A common strategy to promote drug absorption is the use of ocular permeation enhancers, while an innovative approach is the use of polymeric micelles. In the present work, the two mentioned approaches were coupled by conjugating ocular permeation enhancers (PEG_2000_, carnitine, creatine, taurine) to an inulin-based co-polymer (INU-EDA-RA) in order to obtain self-assembling biopolymers with permeation enhancer properties for the hydrophobic drug dexamethasone (DEX). Inulin derivatives were properly synthetized, were found to expose about 2% mol/mol of enhancer molecules in the side chain, and resulted able to self-assemble at various concentrations by varying the pH and the ionic strength of the medium. Moreover, the ability of polymeric micelles to load dexamethasone was demonstrated, and size, mucoadhesiveness, and cytocompatibility against HCE cells were evaluated. Furthermore, the efficacy of the permeation enhancer was evaluated by ex vivo permeation studies to determine the performance of the used enhancers, which resulted in PEG_2000_ > CAR > TAU > CRE, while entrapment ability studies resulted in CAR > TAU > PEG_2000_ > CRE, both for fluorescent-labelled and DEX-loaded micelles. Finally, an increase in terms of calculated Kp and Ac parameters was demonstrated, compared with the values calculated for DEX suspension.

## 1. Introduction

Ocular drug delivery has always been a challenge for pharmaceutical technologists due to the presence of various anatomically and physiologically static and dynamic barriers that strongly limit the absorption of active molecules. Common strategies to improve drug permeability are to combine the active with another specific agent able to increase permeability (absorption or permeation enhancer) or to chemically modify the structure of the active compound. Permeation enhancers are functional excipients that are included in the formulations to improve the absorption of the active molecule. Ocular permeation enhancers are molecules that are able to modify the corneal barrier transiently and reversibly, thus promoting drug entry into the tissue. They could also be food additives and endogenous molecules that are able to interact with their specific transporters. This interaction results in enhanced drug permeation due to tissue response to the stimulus. On the other hand, another traditional successful strategy to overcome poor permeation across the epithelium is to chemically modify the active compound to enable increased passive transport or to employ the specific transporters located in the corneal tissue [1,2]. This is a key point in the development of effective prodrugs. Prodrugs are bioreversible derivatives that undergo an enzymatic and/or chemical transformation in vivo to release the active parent drug, which can then exert the desired pharmacological effect. Prodrugs are generally proposed to improve the physicochemical, biopharmaceutical, or pharmacokinetic properties of the parent active agents. However, prodrugs are classified as novel active ingredients that should be completely studied and characterized [3].

Nonetheless, in the past two decades, novel non-invasive drug delivery technologies specifically designed to overcome the ocular barriers have emerged, including nanoparticles, micelles, dendrimers, microneedles, liposomes, contact lenses, ocular inserts, and nanowafers. The primary aims of any ocular drug delivery system (DDS) are the maintenance of therapeutic drug concentrations at the target site, the reduction in the dosage frequency, the increase in precorneal residence time, and the enhancement of drug bioavailability [3,4,5]. Among the variety of novel useful DDSs, polymeric micelles are particularly promising. Polymeric micelles are colloidal DDSs, ranging in size from 100 to 1000 nm and composed of amphiphilic self-assembling polymers. This type of drug delivery vehicle is characterized by a hydrophilic corona and a hydrophobic inner core that allows hydrophobic drugs to efficiently solubilize, while producing clear aqueous solutions that consequently result in the drugs being suitable for delivery as eye drops. The administration of micelles for topically applied ophthalmic drug delivery displays many advantages, such as increased water solubility of lipophilic molecules, enhanced residence time in the precorneal ocular pocket, increased permeation, and absorption into the deeper ocular tissues. Furthermore, polymeric micelles are generally relatively safe, nontoxic, and able to protect the drug from degradation, allowing sustained release and in some cases overcoming ocular barriers [3,6,7]. Micelles show excellent ocular absorption enhancement properties due to various factors: increase in drug solubility (for hydrophobic molecules), and consequently enhanced amount of readily absorbable active drug; promotion of effective interaction with all the different corneal layers (from the lipid epithelium and endothelium to the hydrophilic collagenous structure of the stroma) due to the amphiphilic nature of the micelle-forming polymer, thus increasing membrane fluidity and permeability; ability of ad hoc designed polymers to target specific transporters and/or reversibly open the tight junction proteins; and enhancement of residence time of the administration surface due to a polymer’s mucoadhesive nature [8,9,10,11,12]. Polymeric micelles should be suitable for a prodrug-like approach, as they might be easily modified to expose functional molecules that could interact with their specific transporters located in the corneal tissue, thus maybe acting as permeation enhancers.

Based on these considerations, the present work aimed to converge the mentioned traditional and innovative approaches to promote drug permeation through the cornea. As the effectiveness of inulin-based (simple or PEGylated) polymeric micelles to act as successful ocular permeation enhancers was already reported [13,14], in this study an improvement in terms of efficacy was proposed by the use of taurine, carnitine, and creatine, as they can interact with their specific transporters located on the corneal surface and probably could produce a permeation enhancer effect. In this view, the innovation of the present study was to propose a prodrug-like approach as well as to evaluate the actual effectiveness of the proposed molecules as permeation enhancers. To this aim, the selected molecules were evaluated both when chemically inserted into the micelle-forming polymeric backbone as well as when co-administered as a solution together with micelle dispersions. Thus, novel inulin-based co-polymers were synthetized, characterized, and employed to prepare micelles. Dexamethasone (DEX) was chosen as the model drug due to its hydrophobicity and its usefulness in the treatment of both anterior and posterior eye segment disorders. DEX exerts complex and multi-factorial anti-inflammatory, anti-edematous, and anti-angiogenetic effects, thus being effective in treating both anterior eye segment inflammatory diseases as well as the main retinal degenerative pathologies. However, it is characterized by unfavorable physicochemical properties. Indeed, it is hydrophobic and thus poorly water-soluble (logP = 1.83). Consequently, it is generally administered as a suspension, even if this may result in ocular irritation and low patient compliance. Recently, many researchers have been working on DEX-loaded polymer micelles for ocular administration with the scope of improving its corneal permeability and promoting its effectiveness [15,16,17]. Thus, in the treatment of a wide range of ocular diseases, DEX is a perfect model of an effective drug that therefore needs to be loaded into a useful carrier system in order to exploit its potential [18,19,20,21,22].

The proposed micelles were then evaluated in terms of particle size, polydispersity index, Z-potential, mucoadhesiveness, cytocompatibility, and ability to cross the ex vivo bovine cornea, thus promoting DEX permeation and penetration into the corneal tissue.

## 2. Materials and Methods

### 2.1. Materials

Inulin (from dahlia tubers), 4-bis-nitrophenyl carbonate (BNPC), ethylenediamine (EDA), *N*-(3-Dimethylaminopropyl)-*N*′-ethylcarbodiimide hydrochloride (EDC·HCl), *N*-idroxysuccinimide (NHS), retinoic Acid (RA), triethylamine (TEA), anhydrous dimethylformamide (DMFa), *O*-[2-(6-Oxocaproylamino)ethyl]-*O*′-methylpolyehtylene glycol 2000 (PEG_2000_), dexamethasone (DEX), taurine, carnitine, creatine, Sephadex G 15 and G 25, pyrene, mucin type III (from porcine stomach); poly acrylic acid (PAA), and NaHCO_3_ were purchased from Sigma-Aldrich. AlexaFluor-NHS_488_ was purchased from Life Technologies. DPBS pH 7.4 buffer solution was prepared by dissolving 9.55 g of Dulbecco’s phosphate buffer saline (Sigma-Aldrich, St. Louis, MO, USA) in 1 L of bidistilled water. This amount of DPBS corresponds to 0.2 g/L of KCl, 0.2 g/L of KH_2_PO_4_, 8 g/L of NaCl, and 1.15 g/L of Na_2_HPO_4_. HEPES buffer solution pH 7.4 simulating ocular fluids was prepared by dissolving 5.96 g of HEPES (Sigma-Aldrich) and 9 g of NaCl (Sigma-Aldrich) in 1 L of bidistilled water and adjusting pH to 7.4 with NaOH 5 M. Human corneal epithelial (HCE) cells were purchased from ScienCell. Keratinocyte serum-free basal medium and its supplements bovine pituitary extract (BPE) and recombinant human epithelial growth factor (EGF) were purchased from Gibco. All solvents and chemicals were of analytical grade and were used without further purification. ^1^H-NMR spectra were recorded using a Bruker Avance II 300 spectrometer operating at 300.12 MHz. Enhanced microwave synthesis was performed by using a CEM Discover Microwave Reactor. Bovine eyes were kindly supplied by the Istituto Zooprofilattico della Sicilia “A. Mirri”, Palermo, Italy, after authorization of the local Department of Veterinary Prevention (ASP-Palermo, Palermo, Italy). All the animal tissues employed in the present study were obtained from the sacrifice of animals intended for human consumption.

### 2.2. Synthesis of the Inulin (INU) Derivatives

#### 2.2.1. INU-EDA

INU-EDA was synthetized via “enhanced microwave synthesis”, following the procedure previously described [23]. Briefly, INU (200 mg) was activated by using 4-BNPC (236 mg) and then left to react with a large excess of EDA (250 μL). The crude product was precipitated in diethyl ether/dichloromethane mixture 2:1 (*v*/*v*) and washed up once with diethyl ether/dichloromethane mixture 2:1 (*v*/*v*) and three times with acetone. The yellow powder obtained was dispersed in bidistilled water, purified by SEC using a Sephadex G15 column, and then freeze-dried. Yield was 100 ± 1%, with reference to the starting inulin amount.

^1^H-NMR (D_2_O, 300 MHz): δ 2.725–3.175 (4H_EDA_, -NH-**CH_2_**-**CH_2_**-NH_2_), 3.611–3.704 (5H_INU_, -**CH_2_**-OH; -**CH**-CH_2_-OH; -C-**CH_2_**-O-), 3.995–4.127 (2H_INU_, -C-**CH**-OH; -**CH**-OH).

#### 2.2.2. INU-EDA-RA

INU-EDA (300 mg) was processed in order to obtain the amphiphilic derivative INU-EDA-RA, as previously reported [13,14,24]. To summarize, RA (78 mg) was activated via EDC·HCl (75 mg)/NHS (45 mg) in a TEA environment and then left to react with INU-EDA overnight in the dark. The obtained copolymer was precipitated in diethyl ether/dichloromethane mixture 2:1 (*v*/*v*) and then washed up three times in the same mixture. Finally, the solid product was dispersed in bidistilled water, purified by dialysis (SpectraPore RC; cut off: 1 kDa), and then freeze-dried. Yield was 85 ± 3%, with reference to the starting INU-EDA.

^1^H-NMR (DMF-d7, 300 MHz): δ 1.227 (6H_RA_, (**CH_3_**)**_2_**-C-), δ 1.670 (2H_RA_, -C-**CH_2_**-CH_2_-CH_2_-), δ 1.806 (2H_RA_, -C-CH_2_-**CH_2_**-CH_2_-), δ 1.909 (3H_RA_, -C-**CH_3_**), δ 2.249 (5H_RA_ -**CH_2_**-C-C-**CH_3_**), δ 2.573 (3H_RA_, -C-**CH_3_**), δ 3.425–3.885 (5H_INU_, -**CH_2_**-OH; -**CH**-CH_2_-OH; -C-**CH_2_**-O-), 4.277–4.448 (2H_INU_, -C-**CH**-OH; -**CH**-OH).

#### 2.2.3. INU-EDA-RA-PEG

INU-EDA-RA-PEG was synthetized as previously reported [14,24]. In brief, INU-EDA-RA (200 mg) was dissolved in bidistilled water (adjusted to pH 6.8 with HCl 0.1 N) and left to react with PEG_2000_-CHO (43.6 mg) overnight at the dark. The crude product was immediately purified by SEC by using a Sephadex G25 column and then freeze-dried. Yield was 97 ± 2%, with reference to the starting INU-EDA-RA.

^1^H-NMR (DMF-d7, 300 MHz): δ 1.246 (6H_RA_, (**CH_3_**)**_2_**-C-), δ 1.678 (2H_RA_, -C-**CH_2_**-CH_2_-CH_2_-), δ 1.809 (2H_RA_, -C-CH_2_-**CH_2_**-CH_2_-), δ 1.930 (3H_RA_, -C-**CH_3_**), δ 2.249 (5H_RA_, -**CH_2_**-C-C-**CH_3_**), δ 2.594 (3H_RA_, -C-**CH_3_**), δ 3.506–3.934 (5H_INU_, -**CH_2_**-OH; -**CH**-CH_2_-OH; -C-**CH_2_**-O-), δ 3.739 (180H_PEG_, **[-CH_2_-CH_2_-]_45_**), δ 4.283–4.471 (2H_INU_, -C-**CH**-OH; -**CH**-OH).

#### 2.2.4. INU-EDA-TAU-RA

The synthesis of INU-EDA-TAU-RA required a multi-step procedure. INU (250 mg; 1.54 mmol) was dissolved in DMF_a_ (4 mL) at 60 °C and activated by 4-BNPC (236 mg; 0.77 mmol) via enhanced microwave synthesis (60 °C; 25 Watt; 1 h). EDA (250 µL; 3.85 mmol) was dissolved in DMF_a_ (1 mL), and the obtained solution was slowly added to the activated INU solution. The reaction was then kept at 25 °C for 1 h under continuous stirring. Afterwards, a previously prepared taurine (9.6 mg; 0.077 mmol) solution in bidistilled water (100 µL) was added to the reaction mixture, and the reaction was kept under constant stirring at 25 °C for 1 h. The crude product was precipitated in acetone and washed up three times with the same solvent. The obtained solid product was dispersed in bidistilled water, purified by SEC using a Sephadex G15 column, and then freeze-dried. Yield was 98 ± 1%, with reference to the starting INU amount.

The synthetic steps are graphically represented in Appendix A.

^1^H-NMR (D_2_O, 300 MHz): δ 2.751–3.200 (4H_EDA_, -NH-**CH_2_-CH_2_**-NH_2_), 2.994 (4H_TAU_, HO_3_S-**CH_2_-CH_2_**-NH_2_), 3.563–3.770 (5H_INU_, -**CH_2_**-OH; -**CH**-CH_2_-OH; -C-**CH_2_**-O-), 4.000–4.133 (2H_INU_, -C-**CH**-OH; -**CH**-OH).

In order to obtain the amphiphilic derivative INU-EDA-TAU-RA, RA (25.7 mg; 0.0855 mmol) was dissolved in DMF_a_ (1 mL) and treated with EDC∙HCl (24.5 mg; 0.128 mmol) and NHS (14.7 mg, 0.128 mmol) in a TEA environment (14 µL) at 25 °C for 4 h in the dark, in order to activate its carboxylic group. Subsequently, the activated RA solution was added to a previously prepared INU-EDA-TAU (100 mg; 0.57 mmol) dispersion in DMF_a_, and the reaction was left at 25 °C overnight in the dark under continuous stirring. The crude product was precipitated in diethyl ether/dichloromethane mixture 2:1 (*v*/*v*) and then washed up three times in the same mixture. Finally, the solid product was dispersed in bidistilled water, then purified by dialysis (SpectraPore RC; cut off: 1 kDa), and then freeze-dried. Yield was 88 ± 4%, with reference to the starting INU-EDA-TAU.

The synthetic steps are graphically represented in Appendix A.

^1^H-NMR (DMF-d7, 300 MHz): δ 1.224 (6H_RA_, (**CH_3_**)**_2_**-C-), δ 1.673 (2H_RA_, -C-**CH_2_**-CH_2_-CH_2_-), δ 1.818 (2H_RA_, -C-CH_2_-**CH_2_**-CH_2_-), δ 1.927 (3H_RA_, -C-**CH_3_**), δ 2.230 (5H_RA_ -**CH_2_**-C-C-**CH_3_**), δ 2.558 (3H_RA_, -C-**CH_3_**), δ 3.939 (5H_INU_, -**CH_2_**-OH; -**CH**-CH_2_-OH; -CH-**CH_2_**-O-), 4.291–4.460 (2H_INU_, -C-**CH**-OH; -**CH**-OH).

#### 2.2.5. INU-EDA-RA-CAR

To obtain INU-EDA-RA-CAR, carnitine (6.1 mg; 0.03815 mmol) was dissolved in bidistilled water (500 µL). Subsequently, EDC·HCl (11 mg; 0.057 mmol) and NHS (6.6 mg; 0.057 mol) were added in order to active its carboxylic group, the pH was adjusted to 5–6 with HCl 0.1 N, and the reaction was kept at 25 °C for 4 h. Afterwards, the reaction mixture was added to a previously prepared INU-EDA-RA (100 mg; 0.545 mmol) dispersion in bidistilled water (2 mL), the pH was adjusted to 4–5 with HCl 0.1 N, and the reaction was kept under constant stirring at 25 °C overnight in the dark. The crude product was immediately purified by dialysis (SpectraPore RC; cut off: 100–500 Da) and then freeze-dried. Yield was 95 ± 3%, with reference to the staring INU-EDA-RA.

The synthetic steps are graphically represented in Appendix A.

^1^H-NMR (DMF-d7, 300 MHz): δ 1.239 (6H_RA_, (**CH_3_**)**_2_**-C-), δ 1.697 (2H_RA_, -C-**CH_2_**-CH_2_-CH_2_-), δ 1.834 (2H_RA_, -C-CH_2_-**CH_2_**-CH_2_-), δ 1.936 (3H_RA_, -C-**CH_3_**), δ 2.027 (2H_CAR_, OH-CH-**CH_2_**-COOH), δ 2.244 (5H_RA_, -**CH_2_**-C-C-**CH_3_**), δ 2.574 (3H_RA_, -C-**CH_3_**), δ 3.132 (9H_CAR_, -N(**CH_3_**)**_3_**), δ 3.929 (5H_INU_, -**CH_2_**-OH; -**CH**-CH_2_-OH; -C-**CH_2_**-O-), 4.282–4.499 (2H_INU_, -C-**CH**-OH; -**CH**-OH).

#### 2.2.6. INU-EDA-RA-CRE

To obtain INU-EDA-RA-CRE, creatine (2.14 mg; 0.01635 mmol) was dissolved in bidistilled water (500 µL) by adding the minimum amount of HCl 0.1 N necessary to obtain a clear solution. Afterwards, EDC·HCl (4.8 mg; 0.025 mmol) and NHS (2.9 mg; 0.025 mol) were added in order to active its carboxylic group. At the same time, INU-EDA-RA (100 mg; 0.545 mmol) was dissolved in 2 mL of bidistilled water and then immediately added to the CRE clear solution. Finally, pH was adjusted to 4–5 with HCl 0.1 N, and the reaction was kept under constant stirring at 25 °C overnight in the dark. The crude product was immediately purified by dialysis (SpectraPore RC; cut off: 100–500 Da) and then freeze-dried. Yield was 92 ± 2%, with reference to the staring INU-EDA-RA.

The synthetic steps are graphically represented in Appendix A.

^1^H-NMR (DMF-d7, 300 MHz): δ 1.224 (6H_RA_, (**CH_3_**)**_2_**-C-), δ 1.661 (2H_RA_, -C-**CH_2_**-CH_2_-CH_2_-), δ 1.794 (2H_RA_, -C-CH_2_-**CH_2_**-CH_2_-), δ 1.903 (3H_RA_, -C-**CH_3_**), δ 2.230 (5H_RA_, -**CH_2_**-C-C-**CH_3_**), δ 2.582 (3H_RA_, -C-**CH_3_**), δ 2.800 (3H_CRE_, -N-**CH_3_**), δ 3.903 (5H_INU_, -**CH_2_**-OH; -**CH**-CH_2_-OH; -C-**CH_2_**-O-), 4.291–4.460 (2H_INU_, -C-**CH**-OH; -**CH**-OH).

### 2.3. Evaluation of Critical Aggregation Concentration (CAC)

The self-assembling ability of the amphiphilic derivatives INU-EDA-RA, INU-EDA-RA-PEG, INU-EDA-RA-CRE, INU-EDA-RA-CAR, and INU-EDA-TAU-RA was evaluated spectrofluorimetrically using a RF-5301 PC Shimadzu spectrofluorimeter and pyrene as probe (emission wavelength of 373 ± 1 nm, excitation wavelength of 333 ± 1 nm). An amount of 10 μL of a pyrene stock solution (6.0 × 10^−5^ M in acetone) was placed into vials and evaporated in an orbital shaker at 37 °C. Aqueous solutions of the amphiphilic derivatives in bidistilled water, DPBS pH 7.4, and HEPES pH 7.4 (concentration range: 4 × 10^−4^–2 mg/mL) were prepared, and then 1 mL of each solution was added to the pyrene residue (pyrene final concentration = 6.0 × 10^−7^ M). The solutions were kept at 37 °C overnight under continuous stirring in order to allow pyrene equilibration with the copolymers. The experiments were performed in triplicate for each aqueous medium.

### 2.4. Preparation of Empty, Fluorescent-Labelled and DEX-Loaded Polymeric Micelles

To prepare both the empty and the drug-loaded polymeric micelles, the film rehydration method was used. An amount of 50 mg of each amphiphilic derivative was dispersed in 3 mL of DMF and then 35 mg of DEX was added. After complete solubilization of the drug, DMF was evaporated using a R-114 Büchi Rotavapor associated with a B-480 Büchi Waterbath. The obtained film was rehydrated with 5 mL of bidistilled water and kept at room temperature under vigorous and continuous stirring for 3 h. The obtained dispersions were purified from the drug’s excess (solid in water and thus present as a suspension) by centrifugation followed by filtration of the collected supernatant through a 5 μm cellulose membrane. The obtained polymeric micelles were then freeze-dried. Yield was 100 ± 1% for empty micelles and 104.3 ± 2.9% for DEX-loaded micelles.

To prepare fluorescent-labelled polymeric micelles, 50 mg of empty micelles was dispersed in 5 mL of NaHCO_3_ buffer solution pH 8.3 (42 mg in 5 mL of bidistilled water) and pH was adjusted to 8.3 with HCl 0.1 N. Subsequently, Alexa Fluor-NHS_488_ (1 mg in 100 μL of DMSO) was added, and the pH was adjusted again to 8.3. The reaction was kept at 25 °C for 1 h under continuous stirring in the dark. The crude product was immediately purified by SEC using a Sephadex G25 column. Yield was ≈ 95 ± 2% in each case, with reference to the starting amount of empty polymeric micelles.

### 2.5. Dynamic Light Scattering (DLS) and Z-Potential Analysis

DLS and Z-potential analysis were performed in bidistilled water, DPBS pH 7.4, and HEPES buffer solution pH 7.4 at 25 °C using a Malvern Zetasizer NanoZS instrument fitted with a 532 nm laser at a fixed scattering angle of 173°. Both empty, DEX-loaded, and fluorescent-labelled polymer micelle dispersions (0.5 mg/mL) were filtered through a 5 μm cellulose membrane and then analyzed. The intensity average hydrodynamic diameter and polydispersity index (PDI) were obtained by cumulants analysis of the correlation function.

### 2.6. Mucoadhesion Studies

The mucoadhesiveness of the proposed micelles was evaluated by the turbidimetric method reported in the literature and by employing the previously prepared empty micelles [25,26]. Transmittance at 500 nm was evaluated over time both for mucin solution (fixed concentration: 1 mg/mL) and mucin/micelles mixtures (final mucin concentration = 1 mg/mL). To evaluate whether mucoadhesion is concentration-dependent, three different polymeric micelles concentrations were tested: 0.5, 1, and 5 mg/mL. Mucin/micelles mixtures were incubated at 37 °C and analyzed at defined time points (0, 30, 60, 180, and 360 min). Each experiment was performed in triplicate in DPBS pH 7.4 and compared with PAA dispersions (at the same three concentrations) employed as a positive mucoadhesive control.

### 2.7. Drug Loading Percentage (DL%) Determination

An amount of 5 mg of DEX-loaded polymeric micelles was dissolved in DMF_a_, filtered through a 0.45 µm nylon membrane, and then brought to volume (5 mL) with fresh DMF_a_. The amount of DEX loaded into micelles was determined by HPCL analysis by using a HPLC Agilent Instrument 1260 Infinity equipped with a Quaternary Pump VL G1311C and DAD detector 1260 VL, 50 µL injector, and a computer integrating apparatus (OpenLAB CDS ChemStation Workstation). Chromatographic separation was achieved on a reversed-phase column Luna Phenomenex C18 (5U, C18, 100A, size 250 × 4.60) and employing a water/acetonitrile (58:42 *v*/*v*) mixture as mobile phase in isocratic condition. The flow rate was set at 0.8 mL/min and the UV wavelength at 246 nm. Under these conditions, the retention time was 8.5 min. Standard curves were used for quantification of integrated areas under the peaks. The calibration curves were performed in the concentration range of 0.1–100 μg/mL by injecting DEX standard solutions in DMF_a_. HPLC data were highly reproducible and linearly related to concentration (R = 0.999). The amount of drug encapsulated in the polymeric micelles was expressed as mg of DEX contained in 100 mg of DEX-loaded micelles.

### 2.8. In Vitro Drug Release Studies

The dialysis method was chosen to evaluate the drug release profile. An appropriate amount of drug-loaded micelles (drug concentration in the donor compartment: 0.500 mg/mL) was dispersed in DPBS pH 7.4 (800 µL) and placed into a dialysis membrane (SpectraPore; cut off 1 kDa) that was immersed in DPBS pH 7.4 (8 mL) and incubated at 37 °C under continuous stirring (100 rpm) in an orbital shaker for 48 h. At selected time intervals, aliquots from the receptor compartment (1 mL) were withdrawn and immediately replaced with an equal volume of fresh DPBS pH 7.4 to maintain sink conditions. Similarly, the drug diffusion profile was evaluated by inserting into the dialysis membrane both a DEX solution at its maximum water solubility (drug concentration: 0.050 mg/mL) or a free drug suspension (drug amount in the donor compartment: 0.500 mg/mL). Each experiment was performed in triplicate, and results were expressed as a percentage amount of drug released (mean ± standard error), with reference to the starting amount of drug loaded into the donor compartment and taking in account the dilution procedure as a function of incubation time. DEX was quantified spectrophotometrically by a 2401 PC Shimadzu Recording Spectrophotometer UV. Standard curves were used for quantification of drug at 240 nm. Calibration curves were performed in the concentration range of 0.25–50 μg/mL. High reproducibility and linearity related to concentration (R = 0.999) were reported. LOD and LOQ values were calculated according to the literature and resulted in values of 0.11 and 0.23 μg/mL, respectively [27].

### 2.9. Stability Studies of DEX-Loaded Micelles

The stability of DEX-loaded micelles was evaluated on the freeze-dried powder stored at room temperature in the dark. At selected time intervals (1, 2, 3, and 6 months), a carefully weighted sample of micelle was dispersed in DPBS pH 7.4 to perform DLS, PDI, and Z-potential analysis, as described above. Moreover, the DEX amount was also quantified over time, as previously reported. Data were compared with those calculated after evaluation of freshly prepared DEX-loaded micelles. Each sample was stored for 6 months [13,14].

### 2.10. In Vitro Biocompatibility Assay

The biocompatibility was assessed both for the prepared amphiphilic INU derivative and the obtained polymeric micelles by MTS assay on human corneal epithelial (HCE) cells. Cells were seeded in a 96-well plate (5000 cells/well) and grown in defined keratinocyte serum-free basal medium supplemented with BPE and EGF, at 37 °C in 5% CO_2_ humidified atmosphere. After 24 h, the medium was replaced with 200 μL of fresh culture medium containing copolymer solutions (5–100 μg/mL) or empty micelles dispersions (150–700 μg/mL). After 1, 4, and 24 h, the medium was replaced with 200 μL of fresh medium containing 20% of MTS solution. Plates were incubated for an additional 2 h at 37 °C. Then, the absorbance at 490 nm was measured using a microplate reader (Multiskan, Thermo Fisher Scientific, Waltham, MA, USA). Results were expressed as cell viability percentage with reference to the viability of untreated cells. Each experiment was performed in triplicate.

### 2.11. Ex Vivo Evaluations

#### 2.11.1. Tissue Preparation

The corneal tissue was withdrawn from bovine eyeballs extracted from freshly slaughtered domestic bovines. All animal specimens were collected by the Istituto Zooprofilattico della Sicilia “A. Mirri” (Palermo, Italy). The obtained entire eyeballs were immersed in DPBS pH 7.4 containing 0.2% of penicillin/streptomycin mixture for 2 h. Furthermore, corneal specimens were isolated by carefully detaching the corneas from the underlying iris manually with the aid of a nipper. Explanted corneas were washed up in HEPES pH 7.4 and immediately used for transcorneal permeation studies.

#### 2.11.2. Set-Up of the Experimental Model

Appropriate sections of cornea were mounted on vertical Franz-type diffusion cells as a two-compartment open model. The corneal tissue was initially inserted between the donor and the acceptor chambers, both filled with 4.5 mL of HEPES buffer solution pH 7.4 (chosen as simulated ocular fluid) and equilibrated at 35 ± 0.1 °C for 10 min [28]. Subsequently, HEPES pH 7.4 was carefully removed from the donor compartment and replaced with 300 µL of diclofenac sodium salt (DICL) or fluorescein sodium salt (FLUO) solutions 1 mg/mL in HEPES pH 7.4, chosen as model molecules. At selected time intervals (15, 30, 60, 90, 120, 150, 180, 240, 300, and 360 min), aliquots (200 μL) were withdrawn from the acceptor chamber and immediately replaced with fresh HEPES pH 7.4 to maintain sink conditions. Each experiment was carried out at 35 ± 0.1 °C for 6 h under gentle and continuous stirring. The FLUO permeation experiments were further conducted in the dark. Each collected sample containing DICL was quantified spectrophotometrically by a 2401 PC Shimadzu Recording Spectrophotometer UV. Standard curves were used for quantification of the drug at 275 nm. Calibration curves were performed in the concentration range of 0.3–500 μg/mL. High reproducibility and linearity related to concentration (R = 0.999) were reported. On the other hand, FLUO was quantified spectrofluorimetrically by using a RF-5301 PC Shimadzu instrument (emission wavelength of 520 ± 1 nm, excitation wavelength of 485 ± 1 nm). Standard solutions in HEPES pH 7.4 were prepared (concentration range: 0.0001–1 μg/mL) and used to construct the calibration curves (R = 0.999). Results are expressed in term of μg/cm^2^ of DICL/FLUO, permeated as a function of incubation time (means ± standard error of 6 repetitions). The experiments were performed according to the literature in order to confirm the suitability of the experimental method and the integrity of the corneal tissue.

#### 2.11.3. Ex Vivo Transcorneal Permeation

Appropriate sections of cornea were mounted on vertical Franz-type diffusion cells as a two-compartment open model. The corneal tissue was initially inserted between the donor and the acceptor chambers, both filled with 4.5 mL of HEPES buffer solution pH 7.4 (chosen as simulated ocular fluid) and equilibrated at 35 ± 0.1 °C for 10 min [28]. Subsequently, HEPES pH 7.4 was carefully removed from the donor compartment and replaced with the sample to be analyzed. At selected time intervals (15, 30, 60, 90, 120, 150, 180, 240, 300, and 360 min), aliquots (200 μL) were withdrawn from the acceptor chamber and immediately replaced with fresh HEPES pH 7.4 to maintain sink conditions. Each experiment was carried out at 35 ± 0.1 °C for 6 h under gentle and continuous stirring. The ex vivo permeation studies were conducted by uploading into the donor chamber the following dispersions: INU-EDA-RA-Alexa Fluor_488_ (10 mg/mL in HEPES pH 7.4) alone or together with a 2% (mol percentage, with reference to the starting molar amount of INU-EDA-RA) solution of PEG_2000_, creatine, carnitine, or taurine; fluorescent-labelled micelles obtained by the copolymers functionalized with the permeation enhancers (10 mg/mL in HEPES pH 7.4); DEX-loaded micelles (both the simple ones and those chemically functionalized with the ocular permeation enhancers) at a fixed DEX starting concentration (500 μg/mL); DEX-loaded INU-EDA-RA micelles at a fixed DEX starting concentration (500 μg/mL) alone or together with a 2% (mol percentage, with reference to the starting amount of INU-EDA-RA) solution of the permeation enhancers; free DEX solutions (50 μg/mL); and free DEX suspension (500 μg/mL). Each collected sample containing DEX was freeze-dried, and the solid residue was treated with 200 μL of DMF to extract and quantify DEX by HPLC analysis, as described above. On the other hand, aliquots from the ex vivo fluorescent-labelled micelles permeation studies were immediately quantified spectrofluorimetrically by using a RF-5301 PC Shimadzu spectrofluorimeter (emission wavelength of 520 ± 1 nm, excitation wavelength of 485 ± 1 nm). To quantify the fluorescent-labelled micelles, standard solutions in HEPES pH 7.4 were prepared (concentration range: 5 × 10^−7^–1 mg/mL) and used to construct the calibration curves (R > 0.990). Results are expressed in term of μg/cm^2^ of DEX/micelles permeated as a function of incubation time (means ± standard error of six repetitions).

#### 2.11.4. Drug Retention in Corneal Tissue

At the end of each permeation experiment, the corneal tissues were collected to quantify the amount of the entrapped DEX/fluorescent-labelled micelles. Corneas were washed with HEPES pH 7.4 to remove any residue on their surface and left in DMF (2 mL) overnight at 37 °C. The extraction liquor was then transferred to a 5 mL flask, brought to volume with fresh DMF, and subjected to HPLC analysis in order to quantify DEX or spectrofluorimetric measurements in order to determine the amount of fluorescent-labelled micelles, as described above. To quantify the fluorescent-labelled micelles, standard solutions in DMF were prepared (concentration range: 5 × 10^−7^–1 mg/mL) and used to construct the calibration curves (R > 0.990). The same procedure was carried out in the absence of samples as well as in the presence of empty micelles used as control to evaluate any eventual interference.

### 2.12. Equations

Drug fluxes (Js) through the corneal tissue were calculated at the steady state per unit area by linear regression analysis of permeation data following the relationship
Js = Q_A_/(A × t) (μg/cm^2^·h^−1^)
where Q_A_ is the amount of DEX/micelles (μg) that passes through a specific area A (1.1304 cm^2^) of the corneal tissue and reaches the acceptor compartment in the time interval t (h). At the steady state, Js is equal to the slope of the straight line obtained. The tlag (min) was determined from the interception of the tangent to the linear portion of the permeation profile with the *x* axis.

The constant of permeability (Kp) was then calculated by the relationship
Kp = Js/Cd (cm/h)
where Cd is the fluorescent-labelled micelles concentration or the drug concentration in the donor compartment (μg/cm^3^), taking into account the total amount of drug actually released from the dialysis bag during 6 h when it is loaded as micelles dispersion (considering results obtained by the in vitro drug release studies) [13,14,29,30].

Similarly, the amount of drug entrapped (De) in the corneal tissue was calculated at the end of each experiment per unit area, following the relationship
De = Q_T_/A (μg/cm^2^)
where Q_T_ is the drug amount (μg) that remains entrapped in a specific area A of the corneal tissue (1.1304 cm^2^).

Accordingly, the accumulation parameter (Ac) was then calculated by the relationship
Ac = De/Cd (cm)

### 2.13. Data Analysis

Data are expressed as mean ±SE. All differences were statistically evaluated by one-way analysis of variance (ANOVA or *F*-test) with the minimum level of significance with *p* < 0.05.

## 3. Results and Discussion

### 3.1. Preparation and Characterization of Micelles

Recently, we demonstrated the potential of inulin-based micelles to overcome ocular barriers [12,13]. In the present new study, the permeation enhancer effect of polymeric inulin-based micelles was further increased by the use of ocular permeation enhancers chemically conjugated to INU-EDA-RA polymer. Inulin (INU) is a natural linear polysaccharide consisting of glucopyranose endcapped fructose units (β-1,2). It is cytocompatible, biodegradable/bioeliminable, water soluble, and has a lot of reactive hydroxyl groups; thus, it is a good candidate with which to produce new derivatives with the appropriate and desired characteristics [14,22,23,24,25]. This attitude was then promoted by partially inserting ethylenediamine (EDA) to introduce primary amine reactive groups and create a stable covalent bond with retinoic acid (RA). This way, an amphiphilic inulin-derivative has already been produced and its effectiveness and usefulness both as drug carrier for hydrophobic molecules and permeation/penetration enhancer has already been proved [13]. Nonetheless, as observed, the chemical insertion of an absorption enhancer molecule (e.g., PEG_2000_) further increases the potential of this type of promising drug delivery system [14]. Indeed, polyethylene glycol plays an important role in the solubilization and permeation processes. Moreover, it was reported that micelle PEG-ylation allowed a low CAC value and consequently increased the stability of the DDS [31,32]. Another promising strategy for improving drug permeation through the highly selective corneal tissue is to employ natural molecules that possess specific transporters on the corneal epithelium, such as taurine, carnitine, and creatine, which could interact with the amino acid transporters or the neurotransmitter transporters SLC6 (prodrug-like strategy) [7]. Therefore, the effectiveness of the mentioned molecules was evaluated not only when chemically linked to the polymeric backbone but also when co-administered with the micellar dispersion. Furthermore, as the PEG-ylation of the nanosystem already gave significant and positive results, this molecule was also included as an enhancer to be evaluated in solution in the present study, although it does not possess specific corneal transporters.

To better compare the proposed novel amphiphilic derivatives, the synthetic steps were carried out with the aim of obtaining a similar molar derivatization degree percentage (DD mol%), in terms of the inserted enhancer molecule, of approximately 2%. ^1^H-NMR spectra (Appendix A) were reported and used to calculate the DD mol% values (Table 1).

Subsequently, the self-assembling ability of the synthetized inulin-derivative was evaluated in various aqueous media (bidistilled water, DPBS pH 7.4, and HEPES pH 7.4) by spectrofluorimetric measurement and using pyrene as a probe [18]. The obtained CAC values are reported in Table 2, together with those previously calculated for INU-EDA-RA and INU-EDA-RA-PEG to allow a better comparison.

As reported, the chemical introduction of the enhancer molecules promoted the dependence of the self-assembling phenomenon as a function of the pH and the ionic strength of the aqueous medium, as previously observed for the PEG-ylated derivative. The dependence of the self-assembling ability as a function of the employed medium was probably due to the chemical structure of the selected enhancer molecules. In particular, taurine is a naturally occurring sulfonic acid, carnitine is a quaternary ammonium compound, and creatine is characterized by a guanidine group and can exist in various tautomers in solution. Each of these groups is ionizable or charged, and this could lead to a pH-dependent behavior. The graphs related to CAC evaluations are reported as Appendix A.

Subsequently, the solvent casting method was selected in order to prepare both empty and drug-loaded homogeneous and reproducible polymeric micelles. Dexamethasone (DEX) was chosen as model of active hydrophobic molecule due to its physicochemical characteristics and its ease of detection by UV-VIS or HPLC analysis, as well as because of its pharmacological activities, which make it a useful therapeutic agent to treat both anterior and posterior eye segment pathologies. Indeed, DEX exerts a complex and multi-factorial anti-inflammatory, anti-edematous, and anti-angiogenetic effect, thus being effective in treating both anterior eye segment inflammatory diseases as well as the main retinal degenerative pathologies [15,16,17]. Nonetheless, with DEX being a lipophilic and thus poorly water-soluble molecule (logP = 1.83), it is generally administered as a suspension. However, eye drops in form of suspensions have many problems, such as non-homogeneity, settling, cake formation, clumping of the suspended particles, resuspendibility, and ocular irritation due to the presence of solid particles [19]. Actually, DEX is a perfect model of a potentially effective drug that therefore needs to be loaded into a useful carrier system in order to exploit its potential.

Furthermore, empty micelles were treated to obtain fluorescent-labeled micelles to be used for the ex vivo studies (see below).

The prepared empty fluorescent-labeled and DEX-loaded micelles were immediately characterized in terms of particle size, PDI, Z-potential (in various aqueous media), and drug loading percentage. Indeed, the corneal absorption of micelles greatly depends on the type of polymer employed as well as on the size and on the surface charge of the prepared DDS [7].

As reported in Table 3 the particle size and the PDI of the prepared enhancer-functionalized micelles do not vary by varying the dispersant medium and particle size and PDI do not increase for the fluorescent-labeled micelles or for the DEX-loaded micelles. These data confirmed the reproducibility of the preparation method and the obtainment of homogeneous samples. In contrast, the already published INU-EDA-RA and INU-EDA-RA-PEG micelles exhibited a similar particle size value when empty (≈200–250 nm), but this value increased (≈300–330 nm) for DEX-loaded INU-EDA-RA and INU-EDA-RA-PEG micelles. This was probably due to the enhanced DL% values, which were 13.00 ± 0.70% and 8.10 ± 0.22% for INU-EDA-RA/DEX and INU-EDA-RA-PEG/DEX micelles, respectively [13,14]. The enhancer-functionalized micelles proposed in this new study resulted in being able to load a significant and detectable amount of DEX (4.10–6.60%), and thus they are suitable for further evaluations and studies. Additionally, it was possible to observe that the Z-potential values obviously vary by varying the dispersant media, as the superficial charge depends on the pH and on the interactions that occur with the buffer’s ions. The variation of the Z-potential, with reference to those calculated for the INU-EDA-RA micelles (e.g., 14.82 ± 2.28 mV in bidistilled water), further confirm the surface functionalization with the selected enhancer molecules and the exposure of different groups on the surface of micelles.

### 3.2. Mucoadhesiveness and Cytocompatibility of Micelles

The effectiveness of a topically applied ophthalmic formulation could be limited by the high tear turnover rate (1 mL/mL), the drug loss due to rapid blinking, the reflex tear production, and the presence of the tear-film barrier. To enhance drug bioavailability, ophthalmic DDSs should possess a higher precorneal residence time. Therefore, it is essential to propose a drug delivery system offering longer retention and a sustained release of the active molecule [33,34,35,36,37]. Thus, the effectiveness of an eye drop formulation, also in terms of permeation/penetration enhancer properties, is strongly related to the retention time on the administration and absorption site. Based on these considerations, it is quite relevant to investigate the mucoadhesiveness of the proposed micelles, which should be administered as eye drops. To perform the mucoadhesion tests, the turbidimetric method was chosen, as its results are useful when investigating the interactions that occur between mucin and an aqueous solution/dispersion. Indeed, a mucoadhesive interaction results in the formation of macroaggregates that can be detected as a decrease in transmittance percentage at 500 nm, with reference to the transmittance of a mucin dispersion under the same experimental conditions. As observable (Figure 1), all the proposed micelles show a mucoadhesive behavior, as confirmed by comparing the graphs related to micelles analysis with that referencing the positive control PAA (Figure 1A). As already reported, both INU-EDA-RA (Figure 1B) and INU-EDA-RA-PEG (Figure 1C) micelles resulted in concentration-dependent but time-independent mucoadhesion. Moreover, the presence of PEG allows a slight increase in the mucoadhesiveness.

INU-EDA-TAU-RA micelles (Figure 1D) show a trend of mucoadhesion as a function of incubation time. This could probably be due to their negative Z-Potential (which depends on the presence of TAU on the micelle surface), making it impossible to create rapid electrostatic interactions with the negatively charged mucin and therefore making it probable that the construction of non-electrostatic bonds requires a longer time to establish. Furthermore, their concentration-dependent behavior results were much more evident than those observed for the INU-EDA-RA and INU-EDA-RA-PEG micelles. Actually, the transmittance percentage at 500 nm evaluated after 6 h of incubation went from 88.80% (concentration 0.5 mg/mL) to 64.55% (1 mg/mL), until reaching the value of 54.73% for the highest concentration tested (5 mg/mL), results of which are comparable with the value obtained for INU-EDA-RA-PEG micelles at the same concentration and at the same time point. The main difference with both INU-EDA-RA and INU-EDA-RA-PEG micelles was therefore attributable to the time required to create a stable mucoadhesive interaction.

Generally, INU-EDA-RA-CAR and INU-EDA-RA-CRE micelles resulted in less mucoadhesion than all the other derivatives and were quite comparable with each other.

Indeed, INU-EDA-RA-CAR micelles (Figure 1E) showed a time-independent but concentration-dependent behavior when considering the 0.5 mg/mL and 1 mg/mL dispersion, while a time-related trend was observable when analyzing the 5 mg/mL dispersion.

In contrast, INU-EDA-RA-CRE micelles (Figure 1F) were always characterized by a decreasing trend over time, but only until reaching a plateau (observed after 1 h of incubation for the lowest concentration and after 3 h incubation for the other two tested concentrations). It should seem that the amount of analyzed nanosystems saturate the amount of available mucin.

In view of the obtained results, a simple and univocal comparison is rather complex. In general, it was possible to affirm that the INU-EDA-RA-CAR and INU-EDA-RA-CRE micelles resulted in less mucoadhesion than the others. However, it was difficult to certainly assert which was the most mucoadhesive nanosystem, as it was not possible to refer only to the final value (6 h) of transmittance percentage because the different behaviors as a function of incubation time should also be considered. In any case, it was certain that all the analyzed samples could interact with a mucin dispersion, and thus they were all mucoadhesive. Furthermore, the highlighted differences in terms of mucoadhesive behavior are an additional proof of the surface modifications due to chemical functionalization. Additionally, it should be considered that these mucoadhesion studies consider only the creation of macroaggregate due to micelle–mucin interaction. Nevertheless, in vivo mucoadhesion is a complex phenomenon, and it is likely to suppose that molecules such as taurine, creatine, or carnitine should allow the creation of additional interactions by also bonding with their specific corneal transporters.

A commercial dosage form might be a powder to be dissolved/dispersed in an appropriate medium. Taking this view, DEX-loaded micelles in the form of freeze-dried powder were stored at room temperature in the dark for 6 months and analyzed in terms of DLS, PDI, Z-potential, and DL%. The results obtained (see Appendix A) confirmed the stability of the prepared freeze-dried powders, which all remained stable over time. This was quite a satisfactory result because it indicates the possibility of easily storing the proposed DEX-loaded micelles, which could be simply resuspended once needed.

Subsequently, the in vitro cytocompatibility against HCE cells was evaluated in order to assess the biocompatibility of the enhancer-functionalized micelles on the ocular surface. Biological evaluations were performed employing both polymeric solutions (polymer concentrations under CAC values) and micellar dispersions (use of empty micelles at concentrations up to CAC values). Results are reported in Figure 2 as percentage of cell viability with reference to untreated cells used as a control. As found, all the proposed polymers and micelles resulted biocompatible at all the tested concentrations and at all the chosen incubation times.

### 3.3. Evaluation of DEX Release and Permeation

Micelles can be used to solubilize primarily hydrophobic drugs, which are otherwise difficult to administer as a compliant clear aqueous solution. Lipophilic drugs can be easily encapsulated in the micellar core, which consists of the hydrophobic parts of the co-polymers. As a consequence, drugs are encapsulated by hydrophobic interactions, Van der Waals interactions, London forces, and hydrogen bonding, which could also contribute to maintaining the drug-loaded DDS integrity. As compared with most other nanosized colloidal drug delivery systems (e.g., nanoparticles, liposomes, dendrimers), micelles can have the highest drug loading efficiency, and this characteristic is perfectly suited for ophthalmic formulation, as the amount of administered drug is maximized when a clear solutions is obtained [7,37]. It is also crucial to the ability of drug-loaded micelles to efficiently release the encapsulated active. Consequently, DEX-loaded micelles were tested for in vitro drug release studies. As the graph in Figure 3 shows, all the nanomicelles were capable of releasing DEX and, consequently, bringing some relevant advantages, e.g., the possibility of administering an enhanced drug dose compared with the solution (because of poor water solubility of DEX) and the possibility of solubilizing and releasing higher amounts of free drug compared with DEX suspension, and therefore increasing the dose actually available for absorption. Indeed, it is important to keep in mind that to evaluate the DEX diffusion profile from a solution, the amount of drug loaded into the dialysis bag was 10 times lower than that loaded when embedded into micelles or as a suspension. In general, it should also be noted that INU-EDA-RA/DEX and INU-EDA-RA-PEG/DEX micelles exhibited the slowest drug release profile. This is not surprising since these nanosystems possess the highest DL% values, and thus they seem to be able to interact firmly and strongly with the chosen drug and, consequently, to retain it longer.

Finally, ex vivo permeation studies were performed as they represent the most important characterization aimed at verifying and evaluating the main goal of the present study. To perform these studies, Franz-type vertical diffusion cells were selected as a two-compartment open model, bovine corneas were used because they result strictly similar to the human corneal tissue, and HEPES buffer solution pH 7.4 was used to simulate the ocular fluids. Indeed, while the literature reports different compositions for preparing a simulated tear fluid, most of them comprise molecules such as albumin, globulin, and enzymes that could compromise the quantitative analysis. As a consequence, HEPES buffer has been proved as a convenient and simple substitute for simulated tear fluid [28,38]. First, the suitability of the proposed experimental method was tested by choosing two model molecules and comparing their permeation results with those reported in the literature. According to Pescina et al. (2015), diclofenac sodium salt (DICL) was selected as model of a molecule able to cross the corneal barrier while fluorescein sodium salt (FLUO) was chosen as model of a poorly permeable molecule. Results are reported as Appendix A in terms of drug permeation profiles (Appendix A) and calculated Kp values, together with those values found in the literature (Appendix A). As can be observed, the obtained data are in accordance with the literature data, thus confirming the effectiveness of the experimental set-up and the integrity of the corneal barrier. Moreover, fluorescein sodium salt is a highly water-soluble dye used in ophthalmology to assess the integrity of the corneal epithelium since it binds damaged epithelium and stroma but not intact epithelium. In this view, the amount of FLUO entrapped in the corneal tissue was also evaluated, and it resulted in 1.20 ± 0.24% of the administered dose, further confirming the integrity of the corneal barrier [39]. As the INU-EDA-RA and INU-EDA-RA-PEG micelles already demonstrated their ability to cross the corneal tissue, the effectiveness of the proposed enhancers (PEG_2000_, taurine, carnitine, creatine) in improving empty micelles permeations/penetration was evaluated both when chemically bonded to micelles as well as when co-administered as a solution together with the micellar dispersion. With this aim, the fluorescent-labeled micelles were used to be able to quantify and evaluate the micelles by themselves. The amount of the enhancer loaded into the donor chamber was kept constant and was the 2% (mol/mol), with reference to the amount of INU-EDA-RA.

Figure 4 illustrates the permeation profiles obtained, while Figure 5 shows the amount of micelles that remains entrapped in the corneal tissue.

As reported, the presence of each permeation enhancer allowed an increase the amount of permeated micelles over time (Figure 4). This was true as when the enhancer was chemically linked as well as when it was co-administered as a solution. The results in terms of enhanced permeability showed that PEG_2000_ > carnitine > taurine > creatine. In each case, a lag time was observable, probably related to the time required by micelles to interact with the tissue and cross all the different layers. Additionally, no great variations can be noted when comparing the same experiment performed with the enhancer-functionalized micelles or the co-administered micelle + enhancer solution. A similar behavior was observable when evaluating the amount of micelles entrapped in the corneal tissue at the end of each permeation experiment (Figure 5). As reported, each enhancer was able to increase the amount of DDS accumulated in the cornea. Nevertheless, in this case, the penetration and accumulation enhancement followed this trend: CAR > TAU > PEG_2000_ > CRE. This generally confirms the limited efficacy of CRE compared with the other selected molecules. Furthermore, the obtained data suggest that the presence of TAU or CAR allowed the creation of a stable interaction with the tissue (high increase in terms of accumulation) that could result in a reservoir effect.

It is important to point out that the proposed micellar systems are not static. Actually, they are quite dynamic structures that are continuously in balance with their unimers. As a consequence, it is impossible to determine whether the whole micellar system or the polymer unimers penetrate and permeate through the cornea. Thus, the terms “permeated micelles” or “accumulated micelles” are used for simplification purposes only. Indeed, the continuous balance between micelles and unimers should not be forgotten as it is quite plausible that the amphiphilic polymeric unimers are actually those responsible for penetration into the tissue, allowing the corneal modifications that lead to the final penetration/permeation enhancer effect. These considerations should also be taken into account when evaluating the DEX ex vivo permeation studies (see below).

Once the ability of micelles to interact, penetrate, and permeate through the corneal tissue by themselves was observed, the same experiments were performed by using the DEX-loaded micelles, also comparing the obtained results with those calculated after administration of a DEX solution or suspension. In particular, Figure 6 illustrates the DEX permeation profiles, Figure 7 shows the amount of DEX accumulated in the corneal tissue, and finally Table 4 reports the calculated parameters, such as the drug flux (Js), the constant of permeability (Kp), the lag time (t_lag_), the drug entrapment (De), and the accumulation (Ac) parameters.

It is important to point out that generally the term *lag time* indicates a time delay between the moment in which the drug is administered and the moment at which it is found in the acceptor chamber. However, as observable for all the reported DEX permeation profiles, the selected drug was already detectable in the acceptor fluid at the first considered time point. Consequently, the calculated t_lag_ is actually the “latency time” necessary for the system to achieve the resulting steady state equilibrium. This latency time is probably attributable to the time required for the drug/micellar dispersion to distribute within the mucosal tissue. Indeed, penetration into the tissue is the first step in the complex and multi-step permeation process.

Generally, it is important to notice that the two different sets of experiments are quite superimposable. Indeed, as the permeation enhancer efficacy again followed the trend PEG_2000_ > CAR > TAU > CRE, while the entrapment ability again resulted in CAR > TAU > PEG_2000_ > CRE, thus confirming in both cases the previously observed result with the fluorescent-labeled micelles. Moreover, the presence of the enhancer (in solution as well as chemically bound) always increased the Kp and Ac values. These are the two main values to be considered as they are in reference to the actual DEX dose. However, for DEX-loaded micelles, the differences observable when employing the enhancer-functionalized micelles or co-administering the micelle dispersion + enhancer solution became more evident in terms of permeation (Figure 6) while still remaining almost absent in terms of drug accumulation in the corneal tissue (Figure 7).

As Table 4 highlights, micelles were able to enhance DEX transcorneal permeation and to promote DEX entrapment in the corneal tissue. The most relevant comparison is that between the DEX-loaded micelles and the DEX suspension (which is the currently administered dosage form). As can be observed, the calculated Kp values resulted in an increase of 1.3 times to 14.3 times, respectively, for the worst (INU-EDA-RA-CRE/DEX) and the best (INU-EDA-RA-PEG/DEX) formulation. While the Ac parameter increased from 2.9 to 7.5 times, respectively, for INU-EDA-RA-CRE/DEX and INU-EDA-RA-CAR/DEX nanomicelles. Generally, all the proposed molecules worked as permeation and penetration enhancers both when inserted into the polymeric backbone as when co-administered as a solution, even if it seems that CAR and CRE were more effective as a solution, while PEG and TAU were more effective as a part of the micelle-forming polymer. In some cases, the lack of significant differences could be due to the steric hindrance of the whole nanosystem, which made it difficult to better interact with the corneal transporters. This issue should maybe be avoided by inserting a spacer portion to better expose the selected functional molecule. Nonetheless, the obtained data are still particularly interesting and promising because the coupling of the traditional (use of a permeation enhancer) and the innovative (use of polymeric micelles) approaches allow an important increase in micelles and DEX absorption through the corneal tissue while, at the same time, assuring a high administrable dose, an increased residence time, and a prolonged drug release profile. Finally, it should be noted that CRE resulted in being the least powerful enhancer among the considered molecules.

## 4. Conclusions

The previously encouraging results obtained for INU-EDA-RA and INU-EDA-RA-PEG micelles as effective innovative DDSs for promoting the permeation of hydrophobic drugs across the corneal tissue have been here coupled with the “permeation enhancer” approach. Taurine, carnitine, and creatine were selected as potential ocular permeation enhancer molecules due to their ability to interact with their specific transporters located on the corneal surface. To merge the conventional and the innovative approaches aimed at improving drug transcorneal absorption, the selected molecules were successfully chemically linked to an INU-EDA-RA backbone, thus obtaining novel inulin-based amphiphilic derivatives. The obtained co-polymers—named INU-EDA-TAU-RA, INU-EDA-RA-CAR and INU-EDA-RA-CRE—were evaluated in terms of DD mol% by ^1^H-NMR analysis, which confirmed their actual functionalization with ≈2% of each selected permeation enhancer. Subsequently, their self-assembling ability was confirmed by spectrofluorimetric analysis, which highlighted the pH- and ionic strength-dependence on the aggregation process. These novel derivatives were then used to prepare empty DEX-loaded and fluorescent-labeled polymeric micelles. All the prepared DDSs that resulted were in the nano-size range (<300 nm) and characterized by different surface charge value, as this parameter depends on the functionalization and thus on the superficial exposure of differently charged groups. Empty micelles were used to assess their mucoadhesive properties, and it emerged that the different functionalization allowed different mucoadhesive behaviors, e.g., INU-EDA-TAU-RA micelles resulted in time-dependent mucoadhesive properties, and INU-EDA-RA-CRE micelles seemed to saturate the available mucin after 1–3 h of incubation. DEX-loaded nanomicelles resulted in being able to load a high amount of drug and to release it almost completely within 48 h, while significantly increasing its aqueous solubility. The key proof of concept of the whole study can be seen in the ex vivo permeation experiments, performed by using Franz-type cells and bovine corneas. These studies confirmed a permeation enhancer efficacy trend (PEG_2000_ > CAR > TAU > CRE) and different entrapment ability behaviors (CAR > TAU > PEG_2000_ > CRE), which confirmed that CRE was the less powerful enhancer among the considered molecules. Nonetheless, the presence of an enhancer molecule always allowed an increase in terms of permeation and tissue accumulation of both the fluorescent micelles and the chosen drug. As an example, the administration of the enhancer-functionalized micelles allowed the calculated Kp values for DEX to increase from 1.3 to 14.3 times and the Ac parameter to increase from 2.9 to 7.5 times, compared with the DEX suspension. Even if in some cases there was a lack of statistically significant differences between the set of experiments performed with the enhancer-functionalized micelles rather than the co-administered micelles dispersion + enhancers solution, the obtained data are still particularly interesting and promising because the coupling of the traditional use of a permeation enhancer with the innovative use of polymeric micelles approaches allowed an important increase in micelles and DEX transcorneal permeation, as well as their entrapment in the corneal tissue, thus possibly resulting in a reservoir effect.

## Figures and Tables

**Figure 1 pharmaceutics-13-01431-f001:**
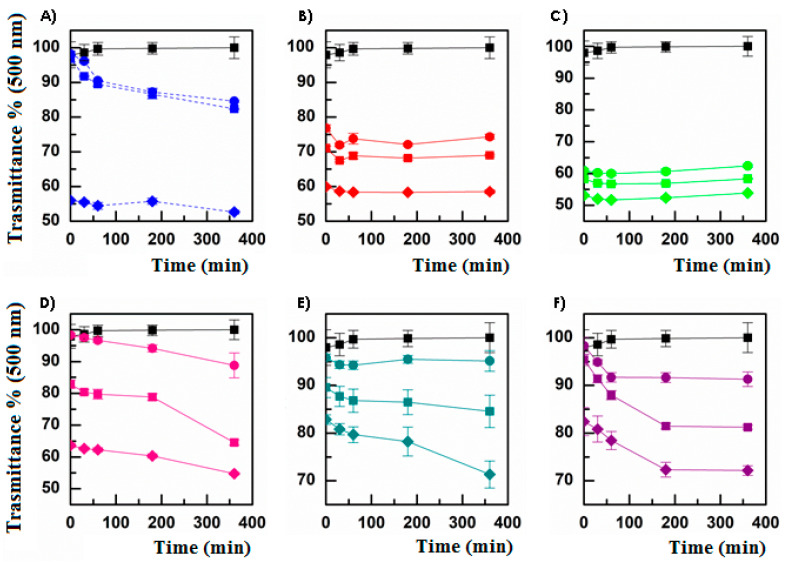
Transmittance percentage at 500 nm of mucin dispersion 1 mg/mL (black), (**A**) PAA (blue); (**B**) INU-EDA-RA micelles (red); (**C**) INU-EDA-RA-PEG micelles (green); (**D**) INU-EDA-TAU-RA micelles (pink); (**E**) INU-EDA-RA-CAR micelles (petrol green); (**F**) INU-EDA-RA-CRE micelles (violet) at the following concentrations: 0.5 mg/mL (circle), 1 mg/mL (square), and 5 mg/mL (rhombus). The obtained values recorded at different concentrations resulted in statistically significant differences (*p* < 0.05).

**Figure 2 pharmaceutics-13-01431-f002:**
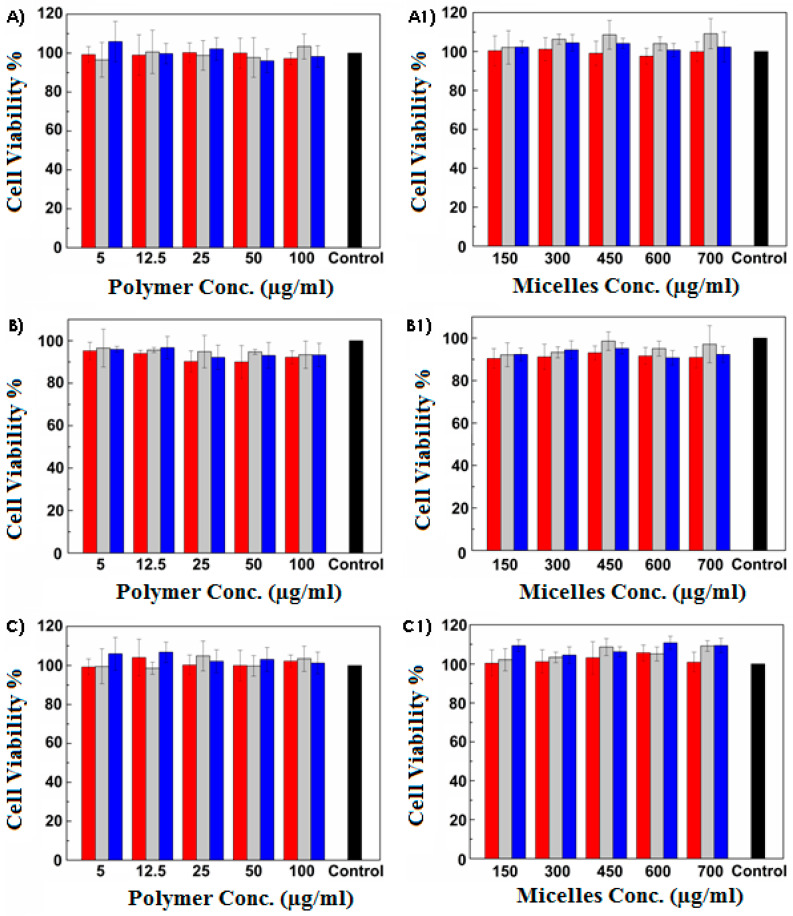
Cytocompatibility studies against HCE cells: Cell viability percentage with reference to untreated cells used as a control (black) after incubation for 1 h (red), 4 h (grey), and 24 h (blue) with INU-EDA-TAU-RA (**A**) solutions and (**A1**) micellar dispersions; INU-EDA- RA-CAR (**B**) solutions and (**B1**) micellar dispersions and INU-EDA-RA-CRE (**C**) solutions and (**C1**) micellar dispersions. The calculated cell viability values at different concentration and different time points did not result in statistically significantly differences (*p* > 0.05).

**Figure 3 pharmaceutics-13-01431-f003:**
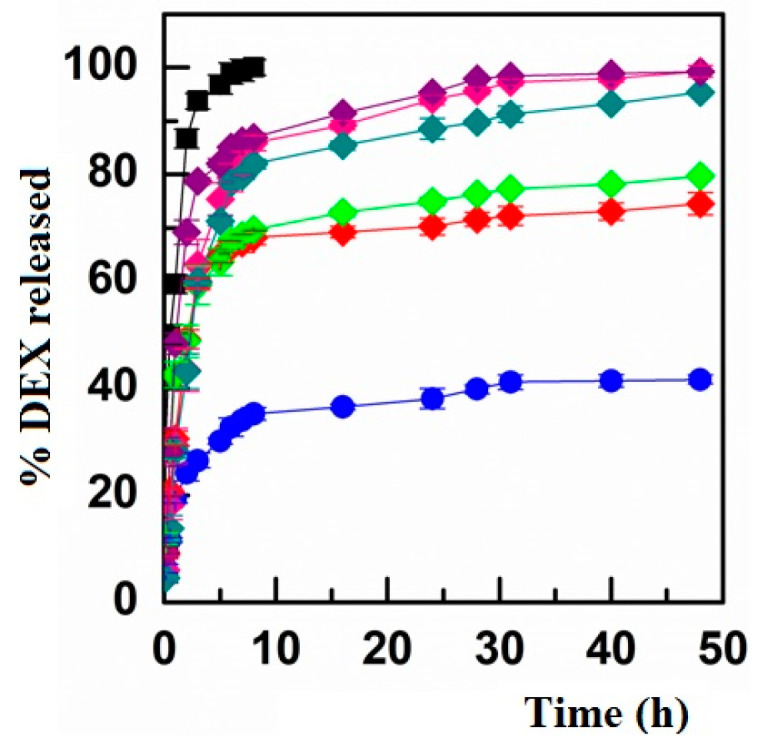
Drug release profiles: percentage of DEX released as a function of incubation time from INU-EDA-RA micelles (red), INU-EDA-RA-PEG micelles (green), INU-EDA-TAU-RA micelles (pink), INU-EDA-RA-CAR micelles (petrol green), and INU-EDA-RA-CRE micelles (violet) compared with the drug diffusion profile from DEX solution (black) and drug dissolution + diffusion profile from DEX suspension (blue). The values recorded when comparing the release behavior of the different micelles resulted in a statistically significant difference (*p* < 0.05).

**Figure 4 pharmaceutics-13-01431-f004:**
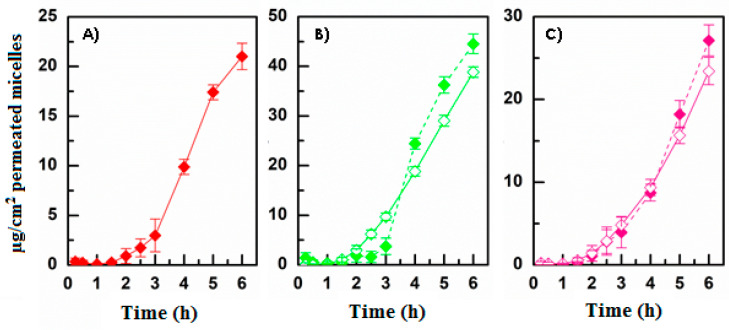
Ex vivo permeation studies: µg/cm^2^ of fluorescent labeled micelles permeated as a function of incubation time after administration of enhancer-functionalized micelles (solid symbols) or co-administration of INU-EDA-RA micelles with the enhancer solutions (open symbols). (**A**) INU-EDA-RA (red), (**B**) INU-EDA-RA-PEG and INU-EDA-RA + PEG solution (green), (**C**) INU-EDA-TAU-RA and INU-EDA-RA + TAU solution (pink), (**D**) INU-EDA-RA-CAR and INU-EDA-RA + CAR solution (petrol green), (**E**) INU-EDA-RA-CRE and INU-EDA-RA + CRE solution (violet). Significance: *p* < 0.05.

**Figure 5 pharmaceutics-13-01431-f005:**
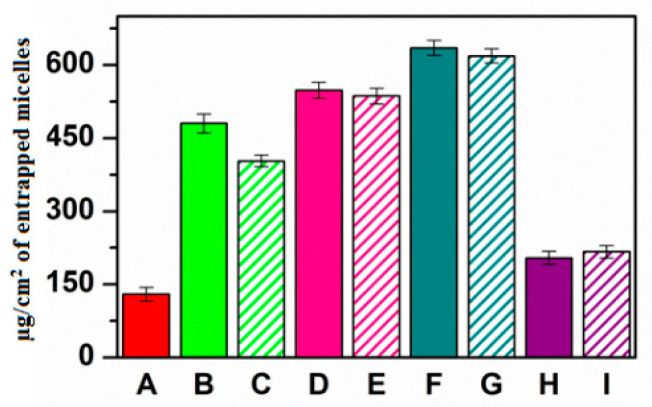
Ex vivo penetration studies: µg/cm^2^ of fluorescent-labeled micelles accumulated in the corneal tissue at the end of each experiment after administration of (A) INU-EDA-RA (red), (B) INU-EDA-RA-PEG (green), (C) INU-EDA-RA + PEG solution (green lines), (D) INU-EDA-TAU-RA (pink), (E) INU-EDA-RA + TAU solution (pink lines), (F) INU-EDA-RA-CAR (petrol green), (G) INU-EDA-RA + CAR solution (petrol green lines), (H) INU-EDA-RA-CRE (violet), (I) INU-EDA-RA + CRE solution (violet lines). Significance: *p* < 0.05.

**Figure 6 pharmaceutics-13-01431-f006:**
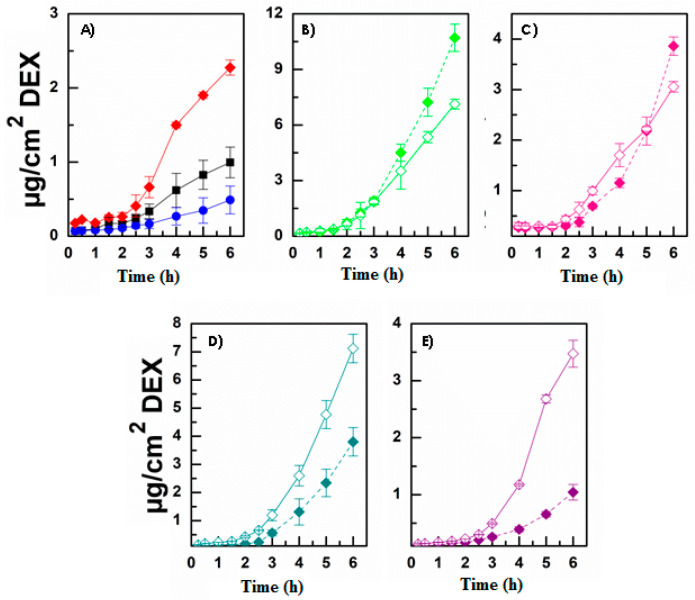
Ex vivo permeation studies: µg/cm^2^ of DEX permeated as a function of incubation time after administration of enhancer-functionalized micelles (solid symbols) or co-administration of INU-EDA-RA micelles with the enhancer solutions (open symbols). (**A**) INU-EDA-RA/DEX (red), DEX solution (black), and DEX suspension (blue), (**B**) INU-EDA-RA-PEG/DEX and INU-EDA-RA/DEX + PEG solution (green), (**C**) INU-EDA-TAU-RA/DEX and INU-EDA-RA/DEX + TAU solution (pink), (**D**) INU-EDA-RA-CAR/DEX and INU-EDA-RA/DEX + CAR solution (petrol green), (**E**) INU-EDA-RA-CRE/DEX and INU-EDA-RA/DEX + CRE solution (violet). Significance: *p* < 0.05.

**Figure 7 pharmaceutics-13-01431-f007:**
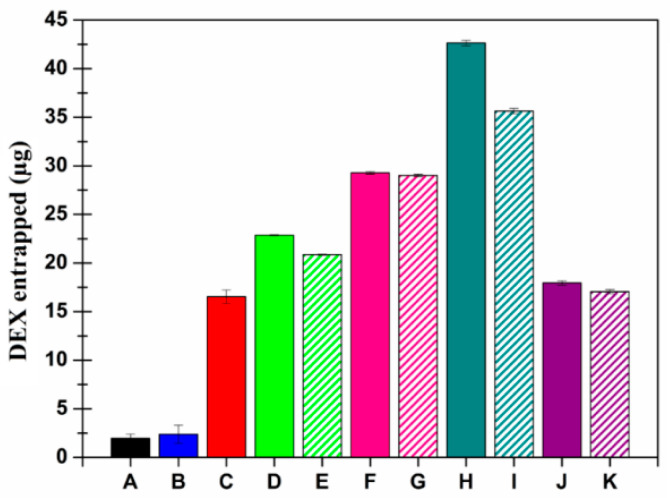
Ex vivo penetration studies: µg of DEX accumulated in the corneal tissue at the end of each experiment after administration of (A) DEX solution (black), (B) DEX suspension (blue), (C) INU-EDA-RA (red), (D) INU-EDA-RA-PEG (green), (E) INU-EDA-RA + PEG solution (green lines), (F) INU-EDA-TAU-RA (pink), (G) INU-EDA-RA + TAU solution (pink lines), (H) INU-EDA-RA-CAR (petrol green), (I) INU-EDA-RA + CAR solution (petrol green lines), (J) INU-EDA-RA-CRE (violet), (K) INU-EDA-RA + CRE solution (violet lines). Significance: *p* < 0.05.

**Table 1 pharmaceutics-13-01431-t001:** DD mol% of EDA, RA, PEG, CAR, CRE, and TAU with reference to INU repeating units and calculated by ^1^H-NMR analysis.

	DD mol% EDA	DD mol% RA	DD mol% PEG	DD mol% CAR	DD mol% CRE	DD mol% TAU
INU-EDA-RA	11.30 ± 1.50	4.30 ± 0.30	-	-	-	-
INU-EDA-RA-PEG	11.30 ± 1.50	4.30 ± 0.30	1.50 ± 0.25	-	-	-
INU-EDA-RA-CAR	11.30 ± 1.50	4.30 ± 0.30	-	2.00 ± 0.13	-	-
INU-EDA-RA-CRE	11.30 ± 1.50	4.30 ± 0.30	-	-	1.90 ± 0.15	-
INU-EDA-TAU-RA	11.30 ± 1.50	4.30 ± 0.30	-	-	-	2.16 ± 0.10

**Table 2 pharmaceutics-13-01431-t002:** Critical aggregation concentration (CAC) values reported in mg/mL ± standard error and calculated both in bidistilled water, DPBS pH 7.4, and HEPES pH 7.4 for each synthetized inulin-based amphiphilic derivative.

	Bidistilled Water (mg/mL)	DPBS pH 7.4 (mg/mL)	HEPES pH 7.4 (mg/mL)
INU-EDA-RA	0.135 ± 0.003	0.136 ± 0.002	0.135 ± 0.003
INU-EDA-RA-PEG	0.209 ± 0.004	0.073 ± 0.001	0.185 ± 0.002
INU-EDA-RA-CAR	0.100 ± 0.002	0.110 ± 0.002	0.140 ± 0.003
INU-EDA-RA-CRE	0.245 ± 0.004	0.150 ± 0.003	0.125 ± 0.003
INU-EDA-TAU-RA	0.142 ± 0.003	0.125 ± 0.002	0.135 ± 0.002

**Table 3 pharmaceutics-13-01431-t003:** Particle size (nm), PDI, Z-potential (mV) e Drug loading % (*w*/*w*) ± standard error evaluated for all the prepared micelles.

	DL% (*w*/*w*)	Particle Size (nm)	PDI	Z-Potential (mV)	Medium
INU-EDA-TAU-RA	−	227.41 ± 12.33	0.159	−7.72 ± 0.88	Water
		225.62 ± 11.70	0.145	−2.65 ± 0.13	DPBS
		234.40 ± 15.62	0.149	−2.96 ± 0.20	HEPES
INU-EDA-TAU-RA-Alexa Fluor_488_	−	222.78 ± 13.81	0.169	−6.99 ± 0.40	Water
		228.98 ± 15.00	0.160	−2.06 ± 0.11	DPBS
		229.99 ± 14.67	0.159	−2.49 ± 0.16	HEPES
INU-EDA-TAU-RA/DEX	6.60 ± 0.56%	226.00 ± 10.42	0.100	−9.10 ± 0.74	Water
		250.63 ± 18.17	0.102	−3.22 ± 0.54	DPBS
		235.00 ± 13.31	0.198	−0.27 ± 0.01	HEPES
INU-EDA-RA-CAR	−	220.22 ± 12.03	0.109	18.72 ± 1.03	Water
		223.94 ± 16.00	0.125	−1.65 ± 0.12	DPBS
		225.49 ± 15.47	0.139	−1.96 ± 0.10	HEPES
INU-EDA-RA-CAR-Alexa Fluor_488_	−	224.54 ± 11.19	0.115	16.99 ± 0.80	Water
		228.89 ± 12.09	0.150	−1.09 ± 0.10	DPBS
		226.97 ± 13.40	0.147	−1.40 ± 0.14	HEPES
INU-EDA-RA-CAR/DEX	4.10 ± 0.33%	224.00 ± 11.87	0.131	20.33 ± 1.13	Water
		230.60 ± 14.14	0.109	−1.22 ± 0.21	DPBS
		231.20 ± 13.99	0.118	−2.27 ± 0.15	HEPES
INU-EDA-RA-CRE	−	227.41 ± 13.09	0.159	12.30 ± 0.94	Water
		225.66 ± 12.00	0.196	−1.65 ± 0.18	DPBS
		234.43 ± 18.01	0.138	−2.96 ± 0.22	HEPES
INU-EDA-RA-CRE-Alexa Fluor_488_	−	229.01 ± 11.50	0.163	11.98 ± 0.99	Water
		228.81 ± 12.09	0.158	−1.49 ± 0.23	DPBS
		232.32 ± 17.00	0.166	−2.50 ± 0.30	HEPES
INU-EDA-RA-CRE/DEX	4.20 ± 0.20%	226.00 ± 17.08	0.140	10.5 ± 0.79	Water
		250.60 ± 19.22	0.191	−3.63 ± 0.33	DPBS
		235.01 ± 13.66	0.108	−4.8 ± 0.35	HEPES

**Table 4 pharmaceutics-13-01431-t004:** Js (µg/cm^2^·h^−1^), Kp (cm/h), De (µg/cm^2^), and Ac (cm) ± standard error calculated by the ex vivo permeation studies after administration of DEX solution, suspension, and DEX-loaded micelles also in presence of the selected permeation enhancers (both chemically bound and as a solution).

	Js (µg/cm^2^·h^−1^)	Kp (cm/h)	t_lag_ (min)	De (µg/cm^2^)	Ac (cm)
DEX solution	0.218 ± 0.009	0.0044 ± 0.0004	90	1.736 ± 0.379	0.0347 ± 0.0012
DEX suspension	0.093 ± 0.005	0.0006 ± 0.0001	90	2.101 ± 0.835	0.0128 ± 0.0014
INU-EDA-RA/DEX	0.535 ± 0.009	0.0016 ± 0.0001	90	14.622 ± 0.616	0.0441 ± 0.0013
INU-EDA-RA-PEG/DEX	2.909 ± 0.010	0.0086 ± 0.0006	138	20.231 ± 0.048	0.0600 ± 0.0015
INU-EDA-RA/DEX + PEG	1.731 ± 0.011	0.0052 ± 0.0004	102	18.455 ± 0.053	0.0556 ± 0.0012
INU-EDA-TAU-RA/DEX	0.953 ± 0.008	0.0024 ± 0.0002	192	25.903 ± 0.120	0.0652 ± 0.0013
INU-EDA-RA/DEX + TAU	0.713 ± 0.008	0.0022 ± 0.0001	100	25.673 ± 0.136	0.0773 ± 0.0014
INU-EDA-RA-CAR/DEX	1.001 ± 0.010	0.0026 ± 0.0001	162	37.722 ± 0.245	0.0961 ± 0.0013
INU-EDA-RA/DEX + CAR	1.994 ± 0.012	0.0060 ± 0.0005	168	31.527 ± 0.233	0.0950 ± 0.0011
INU-EDA-RA-CRE/DEX	0.326 ± 0.003	0.0008 ± 0.0001	156	15.886 ± 0.197	0.0373 ± 0.0010
INU-EDA-RA/DEX + CRE	1.044 ± 0.009	0.0031 ± 0.0002	138	15.090 ± 0.188	0.0455 ± 0.0011

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
