# Peer review of "Inulin-Based Polymeric Micelles Functionalized with Ocular Permeation Enhancers: Improvement of Dexamethasone Permeation/Penetration through Bovine Corneas"

_pharmaceutics, 2021, doi:10.3390/pharmaceutics13091431_

Round 1

Reviewer 1 Report

Line 76: correct 'cornel tissue' and 'permeation enhancer effect'

Line 95: correct 'unfavourable physico-chemical being a lipophilic..'

line 295, also line 365: R should be squared

line 362: correct 'spectrofluorimeter'

please replace the word 'goodness' with e.g. suitability

line 801: 'release up' is not appropriate

Author Response

To the Editor of Pharmaceutics journal

Object: Reviewed Manuscript  (pharmaceutics-1331897) : “Inulin-based polymeric micelles functionalized with ocular permeation enhancers: improvement of dexamethasone permeation/penetration through bovine corneas” Authors: Giulia Di Prima, Mariano Licciardi, Flavia Bongiovì, Giovanna Pitarresi and Gaetano Giammona.

Dear editor,

enclosed please find the reviewed version of the manuscript (pharmaceutics-1331897): “Inulin-based polymeric micelles functionalized with ocular permeation enhancers: improvement of dexamethasone permeation/penetration through bovine corneas” by Giulia Di Prima, Mariano Licciardi, Flavia Bongiovì, Giovanna Pitarresi and Gaetano Giammona., submitted for publication on Pharmaceutics to the following Special Issue " Polymeric Systems to Enhance Drug Penetration and/or Permeation through Biomembranes".

Authors thank the reviewers for their valuable comments and suggestions. Each modification is highlighted in the reviewed version of the manuscript. A point-to-point response to each comment is reported below:

Reviewer 1:

Line 76: correct 'cornel tissue' and 'permeation enhancer effect'

Line 95: correct 'unfavourable physico-chemical being a lipophilic..'

line 362: correct 'spectrofluorimeter'

please replace the word 'goodness' with e.g. suitability

line 801: 'release up' is not appropriate

Authors thank the reviewer. Accordingly, the highlighted mistakes have been corrected.

line 295, also line 365: R should be squared

The reported R value is the pure calculated correlation coefficient and not the R2 value.

Reviewer 2 Report

The resubmitted manuscript (pharmaceutics-1307683) entitled "Inulin-based polymeric micelles functionalized with ocular permeation enhancers: improvement of dexamethasone permeation/penetration through bovine corneas" failed to address previously suggested comments/queries. The manuscript should be majorly revised considering the following suggestions to further improvement in its quality before publication. 

  1. It is suggested to include physicochemical characteristics of dexamethasone which favor/rationalize the formulation design of polymeric micelles to improve its ocular efficacy in the introduction section.  
  2. Many other investigators (like doi: 10.1208/s12249-014-0100-4) have developed dexamethasone-loaded micelles for ocular and other applications. It is suggested to highlight the merits/significance of the current formulation design comparing previously published results in the introduction section. It will be interesting for the reader to understand how to further improve the similar type of drug delivery carrier for the same drug.  
  3. It is suggested to include the LOD and LOQ of the UV method utilized to quantify dexamethasone in in-vitro drug release study. 
  4. Authors are advised to include the optimization results in the preparation of polymeric micelles in section 3.1. in details. The description should also include different variable factors which influence the size and stability of the current formulation development. This section needs to be elaborated according to how the final formulation is ultimately screened and optimized and what are the different factors that influence the robust formulation design in the current investigation. 
  5. Section 2.9, the author has mentioned that “Each sample was stored for more than 6 months”. It is suggested to mention the exact storage period. Mention the guideline according to which this stability study was performed.
  6. It is advised to include the results of the physical and chemical stability of the developed formulation system as supplementary information. Also, include the results of %drug content, size, PDI, and zeta potential during stability study as supplementary information. 
  7.  It is advised to include the powder-XRD results of (i) pure drug – dexamethasone used in investigation (ii) polymer used in the preparation of micelles (iii) freeze-dried blank and drug-loaded micelles.
  8. It is advised to include the image of the finally optimized formulation with figure caption as “Characterization of optimized polymeric micelles (i) Particle size distribution, polydispersity index (PDI) and zeta potential of optimized polymeric micelles (ii) Transmission electron microscopy of an aqueous dispersion of optimized polymeric micelles  (iii) Scanning electron microscopy of freeze-dried blank and drug-loaded optimized polymeric micelles.
  9. Typo/formatting errors in the resubmitted manuscript at Table 4. This might be due to the conversion of the word file to the pdf version. Advised to rectify these formatting errors (section 4. Conclusion, missing/formatting error).  
  10. Determine "lag time" in the ex-vivo permeation study.
  11.  

Author Response

A point-to-point response to each comment of reviewer 2 is reported below:

Reviewer 2:

The resubmitted manuscript (pharmaceutics-1307683) entitled "Inulin-based polymeric micelles functionalized with ocular permeation enhancers: improvement of dexamethasone permeation/penetration through bovine corneas" failed to address previously suggested comments/queries. The manuscript should be majorly revised considering the following suggestions to further improvement in its quality before publication.

  1. It is suggested to include physicochemical characteristics of dexamethasone which favor/rationalize the formulation design of polymeric micelles to improve its ocular efficacy in the introduction section.

The unfavorable DEX characteristics which favor/rationalize its administration as embedded into polymer micelles are related to DEX hydrophobicity and consequently low water-solubility. These lead to the need of administer this drug as a suspension while ocular suspension are not completely suitable in the field of ocular treatment (low patients’ compliance as well as low stability and resuspendibility). As reported in the introduction section (lines 95-98): “However, it is characterized by unfavorable physico-chemical properties. Indeed, it is hydrophobic and thus poorly water-soluble (logP = 1.83). Consequently, it is generally administered as a suspension, even if this may result in ocular irritation and low patients’ compliance.”

  1. Many other investigators (like doi: 10.1208/s12249-014-0100-4) have developed dexamethasone-loaded micelles for ocular and other applications. It is suggested to highlight the merits/significance of the current formulation design comparing previously published results in the introduction section. It will be interesting for the reader to understand how to further improve the similar type of drug delivery carrier for the same drug.

Authors have already inserted in the introduction section some information about the usefulness of DEX in the treatment of many ocular diseases (lines 90-95: “Dexamethasone (DEX) have been chosen as model drug due to its hydrophobicity and to its usefulness in the treatment of both anterior and posterior eye segment disorders. DEX exerts a complex and multi-factorial, anti-inflammatory, anti-edematous and anti-angiogenetic effects thus being effective to treat both anterior eye segment inflammatory diseases as well as the main retinal degenerative pathologies.”) as well as some recent literature about the design of DEX-loaded micelles intended for ophthalmic application (lines 98-102: “Recently, many researchers have been working on DEX-loaded polymer micelles for ocular administration with the scope of improve its corneal permeability and promote its effectiveness [15–17]. Thus, DEX is a perfect model of effective drug in the treatment of a wide range of ocular diseases which therefore needs to be loaded into a useful carrier system in order to exploit its potentiality [18–22].”). However, the crucial aim of the present work is not to produce effective DEX-loaded micelles by themselves to be compared to the previously published ones. The scope of this paper is to demonstrate the usefulness of selected natural molecules (carnitine, creatine and taurine) as ocular permeation enhancers both when chemically linked to micelle surface (prodrug-like approach) as well as when co-administered as a solution. To this purpose, DEX was selected as perfect model molecule due to: i) its hydrophobicity and poor water solubility; ii) its usefulness in the treatment of various ocular diseases and iii) the great recent researchers interest on this molecule. However, it is only a model molecule and this is a key point which makes unnecessary the comparison with other DEX-loaded formulation.

  1. It is suggested to include the LOD and LOQ of the UV method utilized to quantify dexamethasone in in-vitro drug release study.

The required LOD and LOQ values of the UV-VIS method have already been reported in lines 312-317: “DEX was quantified spectrophotometrically by a 2401 PC Shimadzu Recording Spectrophotometer UV. Standard curves were used for quantification of drug at 240 nm. Calibration curves were performed in the concentration range of 0.25–50 µg/ml. It was reported high reproducibility and linearity related to concentration (R= 0.999). LOD and LOQ values were calculated according to the literature and resulted 0.11 and 0.23 µg/ml respectively [28].”.

  1. Authors are advised to include the optimization results in the preparation of polymeric micelles in section 3.1. in details. The description should also include different variable factors which influence the size and stability of the current formulation development. This section needs to be elaborated according to how the final formulation is ultimately screened and optimized and what are the different factors that influence the robust formulation design in the current investigation.

As authors already claimed and as described in the materials and methods section, no different variable factors have been tested. All the obtained micelle samples have been prepared in the same way according to the previously selected parameters (Di Prima et al. 2017 doi: 10.1016/j.ejpb.2017.05.005; Di Prima et al. 2019 doi: 10.1016/j.jddst.2018.10.028). No variable factors have been evaluated in terms of resulting size and stability. As a consequence, no data and no details are missed and both the methods and the results and discussion sections are complete.

  1. Section 2.9, the author has mentioned that “Each sample was stored for more than 6 months”. It is suggested to mention the exact storage period. Mention the guideline according to which this stability study was performed.
  2. It is advised to include the results of the physical and chemical stability of the developed formulation system as supplementary information. Also, include the results of %drug content, size, PDI, and zeta potential during stability study as supplementary information.

Accordingly, stability studies have now been inserted into the revised version of the manuscript. Paragraph 2.9 has been corrected as follow: “The stability of DEX-loaded micelles was evaluated on the freeze-dried powder stored at room temperature in the dark. At selected time intervals (1, 2, 3 and 6 months), a carefully weighted sample of micelle was dispersed in DPBS pH 7.4 to perform DLS, PDI and Z-potential analysis as described above. Moreover, also DEX amount was quantified over time as previously reported. Data were compared to those calculated after evaluation of freshly prepared DEX-loaded micelles. Each sample was stored for 6 month." Results obtained are reported as supplementary materials in Table 1S. Even if micelles were stored for more than 6 months, this was chosen as final time points as a systematic stability study was conducted only up to 6 months.

  1. It is advised to include the powder-XRD results of (i) pure drug – dexamethasone used in investigation (ii) polymer used in the preparation of micelles (iii) freeze-dried blank and drug-loaded micelles.
  2. It is advised to include the image of the finally optimized formulation with figure caption as “Characterization of optimized polymeric micelles (i) Particle size distribution, polydispersity index (PDI) and zeta potential of optimized polymeric micelles (ii) Transmission electron microscopy of an aqueous dispersion of optimized polymeric micelles (iii) Scanning electron microscopy of freeze-dried blank and drug-loaded optimized polymeric micelles.

Authors cannot perform further experiments as the amount of work necessary could require months. Indeed, to prepare novel samples to be further analyzed too many steps are required: i) some materials needs to be bought again; ii) inulin should be dried; ii) all the synthetic steps should be conducted; iv) each polymer should be purified and characterized; v) micelles should be prepared, purified and characterized. Moreover, particle size distribution, PDI and zeta-potential are already reported in table 3 thus a graph should be redundant. Furthermore, we already know that our samples are not suitable for TEM and/or SEM analysis as they are dynamic structure which are continuously in balance with their unimers. Thus it results impossible to obtain good quality images as the polymer unimers constitute the numerically most relevant population and create too much background noise which disturb the analysis. In addition, it should be pointed out that the suggested experiments as well as their possible results will not lead to relevant conclusion regarding the actual scope of the proposed work and will not modify the findings related to our aim.

  1. Typo/formatting errors in the resubmitted manuscript at Table 4. This might be due to the conversion of the word file to the pdf version. Advised to rectify these formatting errors (section 4. Conclusion, missing/formatting error).

Authors thank the reviewer. These errors have now been corrected.

  1. Determine "lag time" in the ex-vivo permeation study.

The lag time values have been calculated and inserted in the revised version of the manuscript (Table 4). The lag time parameter has been inserted in paragraph 2.12. Furthermore, the following explanation has been inserted in the results and discussion section “It is important to point out that generally the term lag time indicates a time delay between the moment in which the drug is administered and the moment in which it is found in the acceptor chamber. However, as observable for all the reported DEX permeation profiles, the selected drug is already detectable in the acceptor fluid at the first considered time point. Consequently, the calculated tlag is actually a 'latency time' necessary for the system to achieve the resulting steady state equilibrium. This latency time is probably attributable to the time required for the drug/micellar dispersion to distribute within the mucosal tissue. Indeed, penetration into the tissue is the first step in the complex and multi-step permeation process.”.

Reviewer 3 Report

The authors have well incorporated all reviewer's comments in the present form of the manuscript.

Author Response

Reviewer 3:

The authors have well incorporated all reviewer's comments in the present form of the manuscript.

Authors thank the reviewer for is appreciation.

Round 2

Reviewer 2 Report

The revised manuscript improved well and try to modify as per the given suggestions. Although some suggestions are not addressed by the authors due to time constrains, indeed the present forms of the manuscript should be considered for publication. I again suggest to incorporate the image of particle size distribution and zeta potential of finally optimized formulation as supplementary information. It will be interesting for the reader. The shape size distribution curve and sometimes small peaks also reveal many things for further interpretation.  Although, author mention in his response, determination of lag time and incorporate it in table 4 and section 2.12 but I not found any such incorporation in revised version. Therefore suggest to rectify this discrepancy.  

Author Response

Authors thank the reviewer and the academic editor for their valuable comments and suggestions. Each modification is highlighted in the reviewed version of the manuscript using the "Track Changes" function in Microsoft Word, so that changes are easily visible to the editors and reviewers. A point-to-point response to each comment is reported below.

We hope that the revised manuscript is now acceptable for publication.

Reviewer 2:

The revised manuscript improved well and try to modify as per the given suggestions. Although some suggestions are not addressed by the authors due to time constrains, indeed the present forms of the manuscript should be considered for publication. I again suggest to incorporate the image of particle size distribution and zeta potential of finally optimized formulation as supplementary information. It will be interesting for the reader. The shape size distribution curve and sometimes small peaks also reveal many things for further interpretation.  Although, author mention in his response, determination of lag time and incorporate it in table 4 and section 2.12 but I not found any such incorporation in revised version. Therefore suggest to rectify this discrepancy.  

Authors thank the reviewer and are really glad to notice that the revised version of the manuscript is suggested to be considered for publication. Regarding the particle size distribution and zeta potential graphs it is authors opinion that the proposed paper already exhibits a large number of images, data and tables and thus the insertion of these graphs (which would be 6 to 10 images) should be too much. Moreover, even if the particle size and zeta potential data are surely indispensable they are not the crucial and key point of the proposed work and consequently the numeric already reported values are enough to support the intended purpose. Regarding the lack of “parts” about the lag time determination it was probably a mistake which is now corrected. The way to determine the lag time is explained in the materials and methods section, data are reported in table 4 (page 23) as follow and a paragraph which explain the calculated t lag values was inserted in the results and discussion section:

Js (µg/cm2·h-1)

Kp (cm/h)

tlag (min)

De (µg/cm2)

Ac (cm)

DEX solution

0.218 ± 0.009

0.0044 ± 0.0004

90

1.736 ± 0.379

0.0347 ± 0.0012

DEX suspension

0.093 ± 0.005

0.0006 ± 0.0001

90

2.101 ± 0.835

0.0128 ± 0.0014

INU-EDA-RA/DEX

0.535 ± 0.009

0.0016 ± 0.0001

90

14.622 ± 0.616

0.0441 ± 0.0013

INU-EDA-RA-PEG/DEX

2.909 ± 0.010

0.0086 ± 0.0006

138

20.231 ± 0.048

0.0600 ± 0.0015

INU-EDA-RA/DEX + PEG

1.731 ± 0.011

0.0052 ± 0.0004

102

18.455 ± 0.053

0.0556 ± 0.0012

INU-EDA-TAU-RA/DEX

0.953 ± 0.008

0.0024 ± 0.0002

192

25.903 ± 0.120

0.0652 ± 0.0013

INU-EDA-RA/DEX + TAU

0.713 ± 0.008

0.0022 ± 0.0001

100

25.673 ± 0.136

0.0773 ± 0.0014

INU-EDA-RA-CAR/DEX

1.001 ± 0.010

0.0026 ± 0.0001

162

37.722 ± 0.245

0.0961 ± 0.0013

INU-EDA-RA/DEX + CAR

1.994 ± 0.012

0.0060 ± 0.0005

168

31.527 ± 0.233

0.0950 ± 0.0011

INU-EDA-RA-CRE/DEX

0.326 ± 0.003

0.0008 ± 0.0001

156

15.886 ± 0.197

0.0373 ± 0.0010

INU-EDA-RA/DEX + CRE

1.044 ± 0.009

0.0031 ± 0.0002

138

15.090 ± 0.188

0.0455 ± 0.0011

“It is important to point out that generally the term lag time indicates a time delay between the moment in which the drug is administered and the moment in which it is found in the acceptor chamber. However, as observable for all the reported DEX perme-ation profiles, the selected drug is already detectable in the acceptor fluid at the first considered time point. Consequently, the calculated tlag is actually a 'latency time' neces-sary for the system to achieve the resulting steady state equilibrium. This latency time is probably attributable to the time required for the drug/micellar dispersion to distribute within the mucosal tissue. Indeed, penetration into the tissue is the first step in the com-plex and multi-step permeation process.”

This manuscript is a resubmission of an earlier submission. The following is a list of the peer review reports and author responses from that submission.

Round 1

Reviewer 1 Report

The proposed study is very comprehensive and with a clear goal.

In this study, the authors investigated the influence of inulin-based polymeric micelles on the improvement of dexamethasone permeation and penetration through bovine corneas.

The manuscript includes interesting information which regards innovative, inulin-based approaches to ameliorate drug permeation through the cornea. Therefore, the authors detailed evaluated novel inulin-based co-polymers that have been synthesized, characterized and employed to prepare micelles.

The special importance of this manuscript is the fact that each novel inulin-based co-polymer evaluated ex vivo in permeation experiments using Franz type cells and bovine corneas. Also, the authors have shown that the presence of the enhancer molecule always allows an increase in terms of permeation and tissue accumulation of both the fluorescent micelles and the chosen drug.

In general, the manuscript is well written and the reviewer found just a few technical errors in the manuscript as follows:

  1. phrases in vitro and ex vivo must put into Italic.
  2. authors need to check English grammar and spelling.

All research activities were performed in detail.

Statistical processing of the obtained results and their categorization were also systematically performed.

Material and methods: Very well and clearly written.

Results: This section is very nicely written with all the necessary and concise accompanying explanations.

The results correspond to the objectives of the study.

The figures are in a satisfactory resolution so that all the mentioned and explained details are clearly visible.

Discussion: The discussion part is explained to a completely satisfactory extent.

The references used are carefully selected and also up-to-date.

Reviewer 2 Report

The manuscript describes the development of functionalized micelles of dexamethasone using permeation enhancers for ocular drug delivery. Authors have conducted a significant amount of work, however there are still several studies missing to support and complement the current findings. In addition the manuscript would significantly benefit from language editing to make the text more readable. Some examples that need revision or rephrasing follow:

Line 30: change to anatomical and physiological

Lines 38-39: ‘allowing to tissue responses resulting in enhanced drug permeation.’

Line 41: ‘take advantages of’

Line 49: ‘polymeric micelles result particularly promising’

Line 54: ‘Topical applications’

Line 58: ‘able to protects’

Line 66: ‘capability of ameliorate’

Line 218: ‘for all medium’

Line 236: ‘was immediately purify’

Line 293: ‘was asses’

Line 395: ‘permeation processed.’

Line 418: ‘As observable’

Line 425:’ Even if the proposed obtained derivatives resulted able to self-aggregate’

Line 450: goodness of the preparation method/ the obtainment of homogeneous samples

Line 456: are anyway able to load

Line 460: interactions which occurs

Line 531: ‘suddenly’ is inappropriate

-Introduction: elaborate more on the literature confirming the use of inulin, taurine, carnitine, and creatine as permeation enhancers in ocular drug delivery

-why was HEPES used instead of the well-established simulated tear fluid composition?

-2.3. Evaluation of Critical Aggregation Concentration (CAC): add excitation and emission slits.same in lines 337-338

-line 229 and 237: yield was >100%/ Yield was ≈ 95%. Please provide SD.

-Line 268: 0.1–0.0001 mg/ml. reverse concentration range and provide it in μg/mL. same in lines 275 and 289.

-2.12. Equations: font size is not the same

-Lines 418-420: provide critical discussion of the results

-For the values reported in table 2 provide the respective graphs (in Supplementary information) from which CAC values were obtained

-TEM analysis should be conducted

--No physicochemical characterizations of the prepared micelles are provided. Authors are encouraged to include such studies (FTIR, DSC etc)

-Cytocompatibility studies have not been performed according to the International Standard (ISO 10993-5) protocol, therefore this indication should be removed from the text. Also authors should have conducted cell viability experiments at a concentration an order of magnitude higher e.g. up to 10 mg/mL similar to the micelle concentration used in the ex vivo studies.

-Figure 4 and 6: use the same scale in the y-axis of all graphs

-Histological studies should be conducted to assess the corneal integrity at the end of the ex vivo experiment and assure that any permeation enhancing effect does not result from tissue integrity compromise

Reviewer 3 Report

The manuscript (pharmaceutics-1307683) entitled "Inulin-based polymeric micelles functionalized with ocular permeation enhancers: improvement of dexamethasone permeation/penetration through bovine corneas" provide a sound description of dexamethasone loaded polymeric micelles for improved ocular delivery with a reasonable set of experimental design. Indeed, the manuscript required revision including following suggestions for further improvement in its quality. 

  1. It is suggested to include physicochemical characteristics of dexamethasone which favor/rationalize the formulation design of polymeric micelles to improve its ocular efficacy in the introduction section.  
  2. Many other investigators (like doi: 10.1208/s12249-014-0100-4 ) have developed dexamethasone-loaded micelles for ocular and other applications. It is suggested to highlight the merits/significance of the current formulation design comparing previously published results in introduction section. It will be interesting for the reader to understand how to further improve the similar type of drug delivery carrier for the same drug.  
  3. It is suggested to merge sections 2.4 and 2.5. as a single section 2.4. with the title "Preparation of placebo, fluorescent labelled, and DEX-loaded polymeric micelles".
  4.  It is suggested to merge sections 2.6, 2.7, and 2.8. as a single section 2.5. with the title "Characterization of DEX-loaded polymeric micelles" and sub-section into 2.5.1, 2.5.2, 2.5.3 accordingly.
  5. It is suggested to include the LOD and LOQ of UV method utilized to quantify dexamethasone in in-vitro drug release study. 
  6. Authors are advised to include the optimization results in the preparation of polymeric micelles in section 3.1. in details. The description should also include different variable factors which influence the size and stability of the current formulation development. This section needs to be elaborated according to how the final formulation is ultimately screened and optimized and what are the different factors that influence the robust formulation design in the current investigation. 
  7. It is advised to include the physical and chemical stability of the developed formulation system. Also, include the results of %drug content, size, PDI, and zeta potential during stability study.